# Continuous-variable protocol for oblivious transfer in the noisy-storage model

Fabian Furrer[1,2], Tobias Gehring [3], Christian Schaffner [4,5], Christoph Pacher [6], Roman Schnabel[7] & Stephanie Wehner[8]

Cryptographic protocols are the backbone of our information society. This includes two-party protocols which offer protection against distrustful players. Such protocols can be built from a basic primitive called oblivious transfer. We present and experimentally demonstrate here a quantum protocol for oblivious transfer for optical continuous-variable systems, and prove its security in the noisy-storage model. This model allows us to establish security by sending more quantum signals than an attacker can reliably store during the protocol. The security proof is based on uncertainty relations which we derive for continuous-variable systems, that differ from the ones used in quantum key distribution. We experimentally demonstrate in a proof-of-principle experiment the proposed oblivious transfer protocol for various channel losses by using entangled two-mode squeezed states measured with balanced homodyne detection. Our work enables the implementation of arbitrary two-party quantum cryptographic protocols with continuous-variable communication systems.

[1] NTT Basic Research Laboratories, NTT Corporation, 3-1 Morinosato-Wakamiya, Atsugi, Kanagawa 243-0198, Japan. [2] Department of Physics, Graduate School of Science, University of Tokyo, 7-3-1 Hongo, Bunkyo-ku, Tokyo 113-0033, Japan. [3] Department of Physics, Technical University of Denmark, Fysikvej, 2800 Kgs Lyngby, Denmark. [4] Institute for Logic, Language and Computation (ILLC) University of Amsterdam, Amsterdam 1098 XG, The Netherlands. [5] QuSoft, Centrum Wiskunde & Informatica (CWI), Amsterdam 1098 XG, The Netherlands. [6] Center for Digital Safety & Security, AIT Austrian Institute of Technology, 1210 Wien, Austria. [7] Institut für Laserphysik und Zentrum für Optische Quantentechnologien, Universität Hamburg, Luruper Chaussee 149, 22761 Hamburg, Germany. [8] QuTech, Delft University of Technology, Lorentzweg 1, 2628 CJ Delft, Netherlands. Correspondence and requests for materials should be addressed to T.G. (email: tobias.gehring@fysik.dtu.dk)

Quantum cryptography can be used to perform cryptographic tasks with information theoretical security based on quantum mechanical principles. Most prominent is quantum key distribution (QKD), which allows to implement a communication link that provides theoretical security against eavesdropping[1–3]. Yet, there are other practically important cryptographic protocols such as oblivious transfer (OT), bit commitment, and secure password-based identification. In these so-called two-party protocols, two distrustful parties (Alice and Bob) engage and want to be ensured that the other party cannot cheat or maliciously influence the outcome. Hence, in contrast to QKD, security for these protocols needs not to be established against an outside attacker but against a distrustful player.

Because of this more demanding security requirement not even quantum physics allows us to implement these tasks securely without additional assumptions[4–10]. An assumption that can be posed on the adversary is to restrict the ability to store information[11,12]. As scalable and long-lived quantum memories are experimentally still very challenging this assumption can easily be justified. In particular, given any constraint on the size of the adversary's storage device, security for two-party protocols can be obtained by sending more signals during the course of the protocol than the storage device is able to handle. This constraint is known as the bounded and more generally noisy-quantum-storage model[13–15].

While OT is the basic building block from which all other two-party protocols can be derived[16], it is possible to use the same techniques to establish security of bit commitment and secure identification. This has been achieved for protocols using a discrete variable (DV) encoding into single photon degrees of freedom (e.g., polarization, path, or time)[17,18]. Using such an encoding OT has been proposed and its security has been studied extensively[13–15,19–22]. Recently, its experimental demonstration has been reported[23].

Here, we propose and experimentally demonstrate in a proof-of-principle experiment an oblivious transfer protocol based on optical continuous-variable (CV) systems. These systems, like classical optical telecommunication systems, encode information into orthogonal quadratures of the electromagnetic field. The similarity to classical telecom systems, room temperature operation, and intrinsic noise filtering by the local oscillator of homodyne detection will allow seamless integration into telecom networks using wavelength division multiplexing to transmit data and perform oblivious transfer or other quantum cryptographic protocols on the same fiber. We prove the security of the protocol in the noisy-quantum-storage model by establishing uncertainty relations, different to the one used in quantum key distribution. The experimental demonstration at a telecommunication wavelength is based on an optical CV setup adapted from a recent implementation of CV QKD[24] which uses entangled two-mode squeezed states and subsequent homodyne measurements in two random orthogonal field quadratures.

## Results

**Oblivious transfer in the noisy-storage model.** In our security proof we derive sufficient conditions for security against a distrustful party having a quantum memory with a bounded classical capacity similar to ref. [19]. The main theoretical ingredients are entropic uncertainty relations for canonically conjugated observables which we derive with and without assumptions on the quantum memory's storage operation and by modeling the quantum memory as bosonic loss channel. While we show that security for arbitrary storage operations is possible, the trade-off in parameters yields very pessimistic rates due to the absence of a tight uncertainty relation. We overcome this problem by assuming that the dishonest party's storage operation is Gaussian.

We consider a one-out-of-two randomized oblivious transfer (1–2 rOT) protocol in which Bob learns one out of two random bit strings. More precisely, Bob chooses a bit $t \in \{0, 1\}$ specifying the bit string he wants to learn, while Alice has no input. Alice's output are two $\ell$-bit strings $\mathbf{s}_0$ and $\mathbf{s}_1$, and Bob obtains an $\ell$-bit string $\widetilde{\mathbf{s}}$. A correct protocol satisfies that the outputs $\mathbf{s}_0$ and $\mathbf{s}_1$ are independent and uniformly distributed, and that Bob learns $\mathbf{s}_t$, i.e., $\widetilde{\mathbf{s}} = \mathbf{s}_t$. To implement 1–2 OT from its randomized version, Alice takes two input strings $\mathbf{x}_0$, $\mathbf{x}_1$ and sends Bob the (bitwise) sums $\mathbf{x}_0 \oplus \mathbf{s}_0$ and $\mathbf{x}_1 \oplus \mathbf{s}_1$ mod 2. Bob can then learn $\mathbf{x}_t$ by adding $\widetilde{\mathbf{s}}$ to $\mathbf{s}_t \oplus \mathbf{x}_t$ (mod 2)[19].

The protocol we propose here to implement 1–2 rOT requires the preparation of Gaussian modulated quadrature squeezed states of light. While indeed the protocol can be implemented using a prepare-and-measure technique, a convenient way to prepare such Gaussian modulated squeezed states is by homodyning one mode of a quadrature entangled two-mode squeezed state—often referred to as EPR state after the authors of their 1935 paper, Einstein, Podolski, and Rosen[25]. Such a state can be generated by mixing two squeezed modes with a balanced beam splitter[26,27]. In the following we will use the entanglement based variant to implement the protocol.

Before Alice and Bob start the actual protocol, they estimate the necessary parameters to run the protocol. The EPR source is located in Alice's lab who is using balanced homodyne detection to estimate the variance of her local thermal state to fix $\alpha_{\text{cut}} > 0$ such that the probability for her to measure a quadrature with an absolute value smaller than $\alpha_{\text{cut}}$ is larger than $p_{\alpha_{\text{cut}}}$ ($p_{\alpha_{\text{cut}}} \approx 1$). Alice and Bob then estimate the correlation coefficient of their measurement outcomes, measured jointly in the same quadrature, to choose an appropriate information reconciliation (IR) code for the protocol. We note that this estimate can be made safely before the protocol even if one of the parties later tries to break the security (see ref. [23] for a discussion).

In the protocol, Alice first distributes $n$ EPR states, each of which is then measured by Alice and Bob who both randomly perform balanced homodyne detection in one of two orthogonal quadratures $X$ and $P$. We assume that Alice and Bob share a phase reference to synchronize their measurements. Alice discretizes the outcomes of the balanced homodyne detection by dividing the range $[-\alpha_{\text{cut}}, \alpha_{\text{cut}}]$ into $2^d$ bins of equal length $\delta$ indexed by $\mathcal{Z} = \{1, \ldots, 2^d\}$. Any measurement lower than $-\alpha_{\text{cut}}$ or larger than $\alpha_{\text{cut}}$ is assigned to the corresponding adjacent bin in $[-\alpha_{\text{cut}}, \alpha_{\text{cut}}]$. Here, it is important that one uses a homodyne detector with subsequent analog-to-digital conversion with a precision larger than $\delta$ and a range larger than $\pm\alpha_{\text{cut}}$. Bob uses the same discretization procedure after scaling his outcomes of the balanced homodyne detection with $1/\sqrt{1 - \mu}$ to account for the losses $\mu$ in the channel. Note that here all transmitted quantum states are used in the protocol, while in the single-photon protocol[18,23] only successful transmissions are back reported. We denote the string of the $n$ discretized outcomes on Alice's and Bob's side as $\mathbf{Z} = (Z_1, \ldots, Z_n)$ and $\mathbf{Y} = (Y_1, \ldots, Y_n)$, respectively.

After completing all the measurements, Alice and Bob wait for a fixed time $\Delta t$. As we will see later, a malicious Bob who wants to cheat has to be able to coherently store the modes in a quantum memory over time $\Delta t$. The rest of the protocol consists of classical post-processing and follows the same idea as the protocol using discrete variables[14,28]. First, Alice sends Bob her basis choices $\boldsymbol{\theta}_A^i$ for each measurement $i = 1, \ldots, n$, that is, whether she measured the quadrature X ($\boldsymbol{\theta}_A^i = 0$) or P ($\boldsymbol{\theta}_A^i = 1$) of the $i$th mode. According to his choice bit $t$, Bob forms the index set $\mathbf{I}_t$ containing all measurements in which both have measured the same quadrature and the complement $\mathbf{I}_{1-t}$ of all measurements in

which they measured different quadratures. Bob then sends the index sets $\mathbf{I}_0$, $\mathbf{I}_1$ to Alice upon which both split their strings of measurement results $\mathbf{Z}$ and $\mathbf{Y}$ into the sub-strings $\mathbf{Z}_k$ and $\mathbf{Y}_k$ corresponding to the indices $\mathbf{I}_k$ ($k = 0, 1$). As elaborated in more detail in the next section, the properties of the EPR source ensure that $\mathbf{Z}_t$ and $\mathbf{Y}_t$ are correlated while $\mathbf{Z}_{1-t}$, $\mathbf{Y}_{1-t}$ are uncorrelated.

Alice then uses a one-way information reconciliation code previously chosen by the two parties and computes syndromes $\mathbf{W}_0$, $\mathbf{W}_1$ for $\mathbf{Z}_0$, $\mathbf{Z}_1$ individually. She then sends $\mathbf{W}_0$, $\mathbf{W}_1$ to Bob, who corrects his strings $\mathbf{Y}_t$ accordingly to obtain $\mathbf{Y}'_t$. The information reconciliation code must be chosen such that up to a small failure probability $\epsilon_{\mathrm{IR}}$ the strings $\mathbf{Z}_t$ and $\mathbf{Y}'_t$ coincide. Finally, Alice draws two random hash functions $f_0$, $f_1$ from a two-universal family of hash functions that map $\mathbf{Z}_0$, $\mathbf{Z}_1$ to $\ell$-bit strings $\mathbf{s}_0$, $\mathbf{s}_1$, respectively. Here, $\ell$ is chosen appropriately to ensure the security of the protocol, see below. Alice then sends Bob a description of $f_0$, $f_1$ and Bob outputs $\widetilde{\mathbf{s}} = f_t(\mathbf{Y}'_t)$.

**Correctness of the 1–2 rOT protocol**. The OT protocol is correct if Bob learns the desired string, i.e. $\mathbf{s}_t = \widetilde{\mathbf{s}}$ and $\mathbf{s}_0$, $\mathbf{s}_1$ are uniformly distributed. The protocol is called $\epsilon_{\mathrm{C}}$-correct if the output distribution of the protocol is $\epsilon_{\mathrm{C}}$-close in statistical distance to the output of a perfect protocol[19]. Thus, $\epsilon_{\mathrm{C}}$ is the failure probability that the protocol is incorrect.

The correctness condition above only has to be satisfied if both parties are honest and follow the rules of the protocol. In that case we can assume that the source and the channel are known. The EPR source has the characteristic property that if both parties measure the X (P) quadrature the outcomes are (anti-)correlated. To turn the anti-correlated outcomes of the P quadrature measurements into correlated ones, Bob simply multiplies his outcomes with $-1$. If Alice and Bob measure in orthogonal quadratures the outcomes are completely uncorrelated. This property of the EPR source implies that the strings $\mathbf{Z}_t$ and $\mathbf{Y}_t$ are correlated while $\mathbf{Z}_{1-t}$, $\mathbf{Y}_{1-t}$ are uncorrelated.

For correctness it is important to demand that the information reconciliation code successfully corrects Bob's string $\mathbf{Y}_t$ with a probability larger than $1 - \epsilon_{\mathrm{IR}}$. Only after successful correction, i.e., $\mathbf{Z}_t = \mathbf{Y}'_t$, it is ensured that $\widetilde{\mathbf{s}} = \mathbf{s}$ after applying the hash function. The properties of the two-universal hash functions also ensure that the outcomes $\mathbf{s}_0$, $\mathbf{s}_1$ are close to uniform. By analyzing the security for Alice we will show that Alice's outcomes are distributed close to uniform even if Bob is dishonest. Thus, if the protocol is $\epsilon_{\mathrm{A}}$-secure for Alice (see next section) our protocol is $\epsilon_{\mathrm{C}}$-correct with $\epsilon_{\mathrm{C}} = \epsilon_{\mathrm{IR}} + 2\epsilon_{\mathrm{A}}$[19,28].

**Security of the 1–2 rOT protocol**. For honest Bob the oblivious transfer protocol is secure if a malicious Alice cannot find out which string $t$ Bob wants to learn. The only information Bob reveals during the entire protocol are the index sets $\mathbf{I}_0$, $\mathbf{I}_1$. However, since honest Bob chooses his measurement basis uniformly at random, the strings $\mathbf{I}_0$, $\mathbf{I}_1$ are completely uncorrelated from $t$. This property implies that the protocol is perfectly secure for Bob without any assumption on the power of Alice. In particular, even if Alice possessed a perfect quantum memory she has no chance to find out $t$.

For honest Alice the oblivious transfer protocol is secure if a malicious Bob can only learn one of the strings $\mathbf{s}_0$, $\mathbf{s}_1$. Similarly to the case of correctness we allow for a small failure probability $\epsilon_{\mathrm{A}}$ that security is not obtained. The precise composable secure definition of the $\epsilon_{\mathrm{A}}$-security for Alice that we employ here is given in terms of the distance to an ideal protocol that is perfectly secure[19].

The security for a honest Alice requires additional assumptions on the power of a malicious Bob to store quantum information. Indeed, it is clear that if a malicious Bob has a perfect quantum memory, he could simply store all the modes until he receives the basis-choice information from Alice. After that he can simply measure all modes in the respective basis such that all the outcomes between Alice and Bob are correlated. This strategy then allows Bob to learn both strings $\mathbf{s}_0$, $\mathbf{s}_1$ and the protocol is completely insecure. But if Bob's quantum storage capacity to store the modes over times longer than $\Delta t$ is limited, he cannot preserve the necessary correlation required to learn both strings. By choosing a sufficiently small output length $\ell$ of the hash function the additional correlation can be erased, and security for Alice can be obtained. The goal of the security proof in this noisy model is to quantify the trade-off between the capability of Bob's quantum memory and the length $\ell$ for which security can be established.

Without restriction of generality we model Bob's available quantum storage ability by $vn$ numbers of channels $\mathcal{F}_{\Delta t}$. Here, the storage rate $v$ relates to the size of the available quantum storage, or also the failure probability to transfer the incoming photonic state successfully into the memory device. Additionally, we allow Bob to apply an encoding operation $\mathcal{E}$ before mapping the incoming mode to the input of his storage device. This encoding map also includes a classical outcome $K$ that can, for instance, result from measuring part of the modes. A schematic of Bob's quantum memory model is illustrated in Fig. 1.

We apply here a similar security proof as the one in ref. [19,28] for discrete variables (see Methods section for details). Therein, the problem of security has been related to the classical capacity $C_{\mathrm{cl}}(\mathcal{F}_{\Delta t})$ of Bob's quantum memory channel $\mathcal{F}_{\Delta t}$. The other important quantity determining the security is the probability with which Bob can correctly guess Alice's discretized measurement outcomes $\mathbf{Z}$ given his classical outcomes of the encoding map and the information of Alice's basis choices. This probability can conveniently be reformulated in terms of the min-entropy

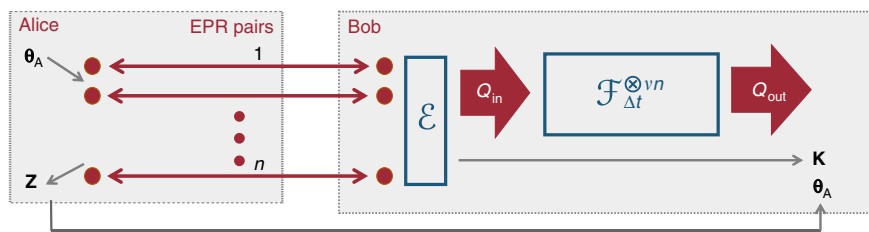

**Fig. 1** The general form of an attack of dishonest Bob. Alice measured her mode of distributed EPR pairs with homodyne quadratures $\boldsymbol{\theta}_{\mathrm{A}}$, yielding (discretized) results denoted $\mathbf{Z}$. Bob's memory attack is modeled by an encoding $\mathcal{E}$ that maps (conditioned on some classical outcome $K$) the $n$ modes to the memory input $Q_{\mathrm{in}}$. The memory $\mathcal{M}$ is modeled by $vn$ uses of the channel $\mathcal{F}_{\Delta t}$. We consider the situations where the encoding $\mathcal{E}$ is arbitrary, a mixture of Gaussian channels or independent and identical over a small numbers of signals $m_{\mathcal{E}}$

which is defined as minus the logarithm of the guessing probability. Furthermore, since we do not require perfect security we use the $\epsilon$-smooth min-entropy $H_{\min}^{\epsilon}(\mathbf{Z}|\boldsymbol{\theta}_A \mathbf{K})$ which is defined as the largest min-entropy optimized over $\epsilon$-close states (see, e.g.,[29]). We emphasize that it is sufficient to condition on the classical information $\boldsymbol{\theta}_A$, $\mathbf{K}$ due to a relation of the smooth min-entropy of all the stored information to the question of how many classical bits can be sent reliably through the storage channel, i.e., $C_{cl}(\mathcal{F}_{\Delta t})$[19] (see Methods section for more details).

A bound on the smooth min-entropy $H_{\min}^{\epsilon}(\mathbf{Z}|\boldsymbol{\theta}_A \mathbf{K})$ is an uncertainty relation. To see this link, we can consider the equivalent scenario in which Bob sends Alice an ensemble of states $\{\rho^k\}$, where $k$ corresponds to the different instances of the random variable $K$. Alice applies on each mode randomly either a discretized $X$ or $P$ measurement. Heisenberg's uncertainty principle tells us that there exists no state for which Bob can correctly guess both outcomes for $X$ and $P$. Since Bob does not know beforehand whether Alice is measuring $X$ or $P$, he will always end up with an uncertainty about Alice's outcomes $\mathbf{Z}$. In the Methods section we derive such uncertainty relations that allow us to bound

$$\frac{1}{n} H_{\min}^{\epsilon}(\mathbf{Z}|\boldsymbol{\theta}_A \mathbf{K}) \geq \lambda^{\epsilon}(\delta, n), \tag{1}$$

with a state-independent lower bound $\lambda^{\epsilon}(\delta, n)$. In the above equation the most crucial difference between the continuous- and the discrete-variable implementation appears. Indeed, while for discrete variables an uncertainty relation for BB84 measurements is required, we here need one for discretized position and momentum observables with finite binning $\delta$.

We have now all ingredients to state the final results. Let us assume that the reliable communication rate of Bob's quantum memory channel decreases exponentially if a coding rate above the classical capacity $C_{cl}(\mathcal{F}_{\Delta t})$ is used. Then, given that $\lambda^{\epsilon}$ satisfies Eq. (1), we obtain an $\epsilon_A$-secure 1–2 rOT if the length of the output bit string is chosen as

$$\ell \leq \frac{n}{2}\left(\lambda^{\mathcal{O}(\epsilon_A)}(\delta, n) - r_{IR} - \nu C_{cl}(\mathcal{F}_{\Delta t})\right) - \mathcal{O}\left(\log \frac{1}{\epsilon_A}\right). \tag{2}$$

Here, $r_{IR} = (1/n)\log|\mathbf{W}_0 \mathbf{W}_1|$ is the rate of bits used for information reconciliation. The explicit dependence on $\epsilon_A$ and the relation between the security and the classical capacity $C_{cl}(\mathcal{F}_{\Delta t})$ are given in the Methods section. If the right hand side of Eq. (2) is negative, security for Alice is not possible.

We see that security can be achieved for sufficiently large $n$ if $\lambda^{\mathcal{O}(\epsilon_A)} - r_{IR} - \nu C_{cl}(\mathcal{F}_{\Delta t})$ is strictly larger than 0. In other words, we need that the uncertainty generated by Alice's measurements should be larger than the sum of the leaked information during information reconciliation and the storage capacity of Bob. It is thus essential to find a tight uncertainty relation Eq. (1). We derive such an uncertainty relation in the Methods section. It turns out that it is difficult to derive a tight bound without further assumptions. This is partly due to the fact that no non-trivial uncertainty relation exists for continuous $X$ and $P$ measurements, i.e., if $\delta$ goes to 0. The uncertainty relation has thus to be derived directly for the discretized $X$ and $P$ measurements. We therefore also derive uncertainty relations under different assumptions on Bob's encoding operation $\mathcal{E}$, namely, under the assumption that the encoding operation is a Gaussian operation and under the assumption that the encoding operation acts independent and identically (i.i.) on a limited number of modes $m_{\mathcal{E}}$. For the explicit form of the uncertainty relations, we refer to the Methods section.

**Security for realistic memory devices.** Let us analyze the security in the case that Bob's quantum memory can be modeled by a

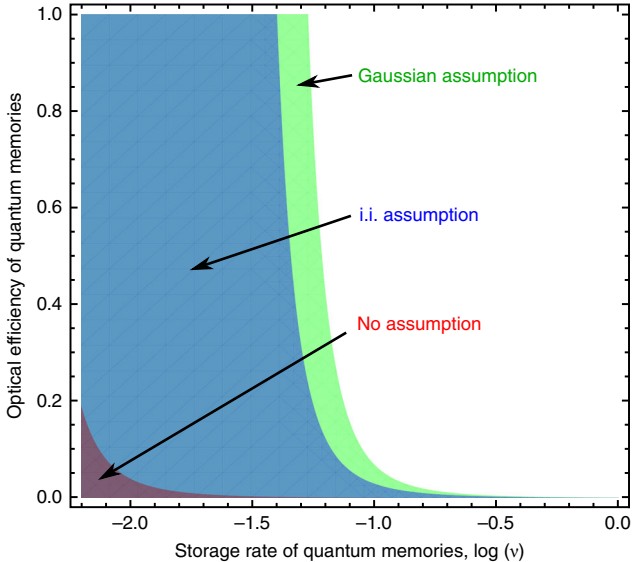

**Fig. 2** Oblivious transfer security regions. The secure regions are obtained for different assumptions imposed on the encoding operation of malicious Bob's quantum memories. We plot optical efficiency $\eta$ of the quantum memories versus the logarithm to basis 10 of the quantum memory storage rate $\nu$. Security is obtained for all values of $\nu$ and $\eta$ marked by the colored regions. The green region is obtained under the assumption that the encoding is Gaussian ($n = 2 \times 10^5$, $\beta = 0.944$, $\delta = 0.1$), the blue region under the assumption that the encoding is independent and identical over at most $m_{\mathcal{E}} = 10$ modes ($n = 10^8$, $\beta = 0.944$, $\delta = 0.1$), and the red region without any assumption, i.e. arbitrary encodings ($n = 10^8$, $\beta = 0.98$, $\delta = 1.0$). The plots are obtained for an EPR source with two-mode squeezing of 12 dB and losses on Alice's and Bob's side of 3 and 6%, respectively. Further parameters: $\epsilon_A = 10^{-7}$, $\alpha_{cut} = 51.2$ and Bob's maximal photon number in the encoding is assumed to be smaller than 100

lossy bosonic channel $\mathcal{N}_n$, where $\eta$ denotes the transmissivity. The classical capacity of this channel has only recently been determined after settling the minimal output entropy conjecture[30,31]. If the average photon number of each code word is smaller than $N_{av}$, it is given by $g(\eta N_{av})$, where $g(x) = (x + 1)g(x + 1) \log_2(x + 1) - x \log_2 x$. An energy constraint is necessary as otherwise the capacity is unconstrained due to a memory that is infinite dimensional.

Recall that we further require that the success probability for reliable communication must drop exponentially to apply the security proof. It has been shown that for this to be the case a constraint on the average number of the photons is not sufficient but one has to impose that every code word is with high probability contained in a subspace with maximally $N_{max}$ photons[32]. Under this maximal photon number constraint the reliable communication vanishes exponentially at a rate above the classical capacity $g(\eta N_{max})$[32–34], so that we can apply our security proof with $C_{cl}(\mathcal{F}_{\Delta t}) = g(\eta N_{max})$.

We plot in Fig. 2 under which assumptions on Bob's quantum storage device security can be obtained. In particular, we consider the situation of arbitrary encoding operations, the situation that Bob's encoding operation is a Gaussian operation, and the situation that Bob's encoding operation is independent and identical over blocks of at most 10 modes. To obtain security, i.e. a positive OT rate, for arbitrary encoding operations, it is necessary to have an information reconciliation code with almost perfect efficiency $\beta = 1$. The information reconciliation efficiency describes the classical communication rate compared to the

asymptotic optimal value, where the latter is achieved for $\beta = 1$. Current codes for CV systems can reach about $\beta = 0.98$[35,36]. The weakest requirements on the parameters have to be imposed under the Gaussian assumption in which security can already be obtained for low numbers of signals $n = 10^5$ (see Methods section). Under the independent and identical encoding assumption, larger numbers of transmitted signals $n = 10^8$ are required to obtain security under similar conditions as in the case of Gaussian operations.

In general, to obtain security a transmittance of the channel between Alice and Bob larger than 0.5 and non-trivial squeezing is required. This result is easily obtained if one takes the asymptotic limit for $n$ to infinity under Gaussian or the identical and independent encoding operations. We note that the identical and independent assumption is no restriction of generality any more in the asymptotic limit[37].

**Experimental demonstration of 1–2 rOT.** We performed a proof-of-principle experimental demonstration of the 1–2 rOT protocol using the experimental setup employed for CV QKD in ref. [24] and sketched in Fig. 3a. The EPR source was located at Alice's location and consisted of two independent squeezed-light sources each producing continuous-wave squeezed vacuum states at 1550 nm by parametric down-conversion[27]. Both states were interfered at a balanced beam splitter with a relative phase of $\pi/2$ thereby exhibiting more than 10 dB entanglement according to the criterion from Duan et al.[38]. Alice kept one of the entangled modes and performed balanced homodyne detection using a low-noise, high quantum efficiency homodyne detector (see details in the Methods section). The homodyned quadrature amplitude was chosen randomly according to random bits generated by a quantum random-number generator based on homodyne measurements on vacuum states. The other entangled mode was sent to Bob via a free-space channel along with a bright local oscillator beam which served as phase reference. Optical loss in this channel was introduced by a variable beam splitter comprising a half-wave plate and a polarizing beam splitter. Bob performed balanced homodyne detection on his mode with a random quadrature chosen by a similar quantum random-number generator. The measurement repetition rate of the system was 100 kHz. For more experimental details we refer to the Methods and the ref. [24].

The classical post-processing was implemented as described above. We chose the number of exchanged signals to be $2.03 \times 10^5$ such that the number of measurement results where both parties have measured in the same bases and where both parties have measured in different bases are both larger than $10^5$ with high probability. We then chose from each set the first $10^5$ for post-processing (i.e., $n = 2 \times 10^5$) to keep the block size of the information reconciliation code constant. From a security perspective this is possible because the size of the set is determined beforehand as part of the protocol. Because the honest player chooses his/her basis string uniformly at random, the choice of these sets is thus out of control of any dishonest player. For the discretization of the measurement outcomes, we used $\alpha_{\text{cut}} = 51.2$ and $\delta = 0.1$, obtaining symbols from an alphabet of size 1024 corresponding to 10 bits per symbol.

The most challenging part is the information reconciliation for which we used a similar strategy as in ref. [24] and detailed in ref. [36]. Here, Alice first communicated the four least significant bits of each symbol in plain to Bob. To correct the remaining 6 bits, she then used a non-binary low-density parity-check (LDPC) code with field size 64 and a code rate $R$ compatible with the estimate of the correlation coefficient $\rho$ from the CM. After Bob has received the syndrome corresponding to his input bit $t$ (ignoring the data corresponding to bit $1 - t$) he ran

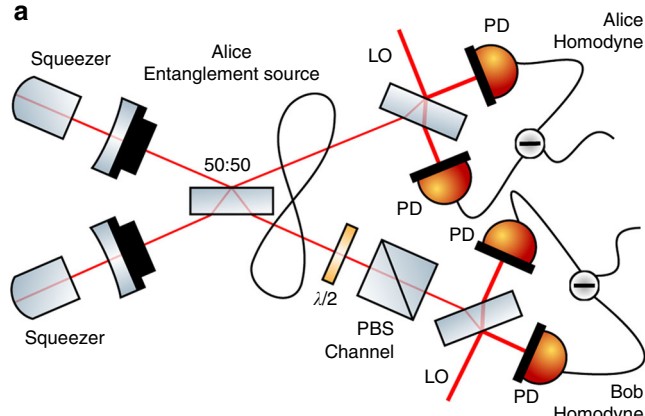

**a**

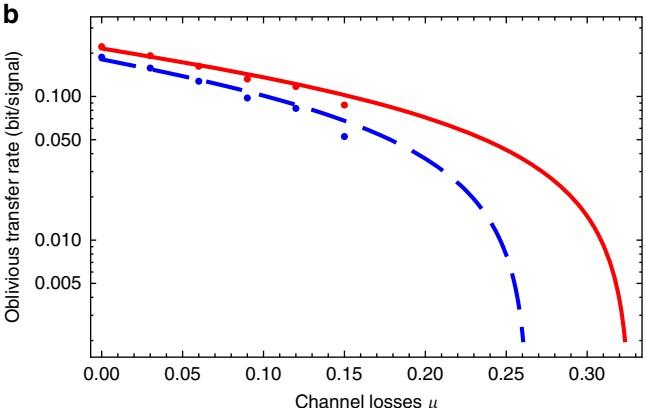

**b**

**Fig. 3** Experimental setup and results. **a** Squeezed light at 1550 nm was generated in two parametric down-conversion sources and superimposed at a 50:50 beam splitter to obtain entanglement. One mode was kept locally by Alice and measured with homodyne detection randomly in the amplitude and phase quadrature. The other mode was sent through a free-space channel simulated by a half-waveplate and a polarizing beam splitter (PBS). Bob then performed homodyne detection randomly in amplitude and phase quadrature. PD photodiode, LO local oscillator. **b** Secure oblivious transfer rate per signal obtained in the experiment. Points correspond to the generated oblivious transfer rates in the experiment for two different storage rates, $\nu = 0.001$ (red) and $\nu = 0.01$ (blue), for quantum memories with a transmittance of 0.75. The lines show simulated oblivious transfer rates obtained by applying a one-sided loss channel with losses $\mu$ to the estimated two-mode squeezed state in the experiment. Parameters: $n = 2 \times 10^5$, $\alpha_{\text{cut}} = 51.2$, $\delta = 0.1$, $\epsilon_A = 10^{-7}$, and Bob's maximal photon number in the encoding is assumed to be smaller than 100

a belief propagation algorithm to correct $Y_t$. In Table 1 we summarize the used code-rates for the different loss scenarios in our experiment.

As family of two-universal hashing functions we selected the mapping of the binary input string to the binary output string by multiplying the input string with a uniformly randomly chosen binary Toeplitz matrix $T$. Multiplication by a Toeplitz matrix is equivalent to linear cross-correlation. This allowed us to make use of the number-theoretic transform to obtain an implementation with computational complexity $O(n \log n)$ and without floating point errors. The binary input strings had a total length of $10^6$ bits (consisting of $n/2 = 10^5$ symbols with 10 bits per symbol). The binary output strings had a length of $\lfloor \ell \rfloor$. Thus the size of $T$ was $\lfloor \ell \rfloor \times 10^6$. The seed (the values for the first row and first column of the Toeplitz matrix) was generated with the quantum random number generator.

**Table 1 Parameters for information reconciliation and privacy amplification**

| Loss | $\sigma_A$ | $\rho$ | capacity | $R$ | $r_{IR}$ | $\beta$ | FER | $\ell$ (kbit) |
|---|---|---|---|---|---|---|---|---|
| 0 | 4.838 | 0.9960 | 3.486 | 0.94 | 4.36 | 0.942 | 0/985 | 44.4 |
| 3% | 4.238 | 0.9936 | 3.151 | 0.92 | 4.48 | 0.943 | 0/1083 | 38.4 |
| 6% | 4.535 | 0.9932 | 3.101 | 0.90 | 4.60 | 0.951 | 0/985 | 32.4 |
| 9% | 4.556 | 0.9923 | 3.013 | 0.88 | 4.66 | 0.941 | 1/1182 | 26.4 |
| 12% | 4.637 | 0.9916 | 2.950 | 0.87 | 4.78 | 0.950 | 0/1083 | 23.4 |
| 15% | 4.584 | 0.9903 | 2.846 | 0.85 | 4.90 | 0.937 | 0/1358 | 17.4 |

Mean values for channel loss, standard deviation of Alice's data $\sigma_A$, correlation coefficient $\rho$, channel capacity, code rate $R$ of LDPC codes over GF(64) used, corresponding leakage rate $r_{IR}$, efficiency $\beta$, frame error rate (FER) and exemplified output length $\ell$ for $\nu = 0.001$ and the other parameters as described in the caption of Fig. 3

The correctness parameter $\epsilon_C = \epsilon_{IR} + 2\epsilon_A$ of the protocol depends on security parameter $\epsilon_A$, which we chose to be $10^{-7}$, and the probability of successful information reconciliation. From the given frame error rates in Table 1, we deduce a success rate of larger than 99.9%, i.e. $\epsilon_{IR} = 10^{-3}$, limited by the amount of experimental data taken, which yields $\epsilon_C = 10^{-3}$. The single frame error for 9% channel loss is thereby due to an error which prevents convergence of the LDPC decoder. The average overall efficiency of the information reconciliation was 94.4%. While generally possible, the temporal drift of the experimental setup in combination with the requirement of achieving a low frame error rate prevented a higher efficiency.

The results are shown in Fig. 3b. We computed the security under the Gaussian assumption and under the assumption on Bob's quantum memories that the maximal photon number in the encoding is smaller than 100. The points correspond to the experimental implementation and the theoretical lines were computed using the estimated CM and the efficiency of the information reconciliation protocol used with the lossless channel. We see that for low channel loss, rates in the order of 0.1 bit per transmitted quantum state are possible. The maximal tolerated loss in the communication channel heavily relies on the assumptions on malicious Bob's storage rate, which we set to $\nu = 0.01$ and $\nu = 0.001$ in Fig. 3b.

**Discussion**

We presented and experimentally demonstrated a protocol for oblivious transfer using optical continuous-variable systems, and showed security against a malicious party with an highly imperfect quantum storage device. For the implementation we used a strongly entangled two-mode squeezed continuous-wave light source, and balanced homodyne detection together with a quantum random-number generator for the measurements. While the employed EPR entangled state was close to optimal in the investigated regime of up to 15% channel loss, security can also be obtained with weaker entangled sources, e.g. for a quantum memory storage rate of $\nu = 0.001$ only about 4 dB of squeezing are necessary to obtain security for 15% loss. More details can be found in the Methods section.

The secure bit rate of the OT protocol is in trade-off with assumptions on the quantum storage device of a dishonest party. In particular, it depends on the classical capacity of the storage device $\mathcal{C}_{cl}$ and the storage rate $\nu$. The storage rate is determined by the size of the available quantum storage and the success rate for transferring the photonic state into the quantum memory. To obtain security for any storage size, one can simply increase the number of signals sent during the protocol. The classical capacity is determined by the efficiency of the quantum memory for writing, storing (over time $\Delta t$) and reading out. Typical storage times of state-of-the-art quantum memories are milliseconds to seconds with some going up to minutes[39].

For low channel losses we obtain rates that are about a factor three larger than those achievable in a previous DV implementation[23] while using significantly smaller block sizes of about $10^5$ compared to $10^7$. However, our implementation is susceptible to losses and requires the optical loss to be generally less than 50%. This limit is a consequence of the analysis we employ in the security proof and is not a fundamental property of CV oblivious transfer. For practical purposes we encounter, however, an even lower loss threshold. For instance, in our experiment losses below 26% for $\nu = 0.01$ and 32% for $\nu = 0.001$ are necessary (see Fig. 3). This allows for an implementation of the protocol in short-range applications like a short free-space link with high collection efficiency, e.g., at an ATM, or a short fiber link of maybe 3–4 km within a business district of a city. Here, we assumed a free-space to fiber coupling efficiency of 95% (achievable with anti-reflex coated fibers), a realistic fiber transmission loss of 0.3 dB/km at 1550 nm and a high efficiency free-space homodyne receiver as implemented in our experiment.

Information reconciliation is required to correct the discretized (non-binary) data. In contrast to the case of DV, where conditioned on the arrival of a photon the bit-error rate is rather low, we require efficient information reconciliation for non-binary alphabets with high probability of success, i.e. low block error rate, since a two-way check ensuring that information reconciliation was successful will in contrast to QKD compromise security.

The security proof presented here can be adapted to other two-party cryptographic protocols such as bit commitment and secure identification using similar ideas and protocols as in[13,18,19,28]. Moreover, the security proof can be refined in various directions. Firstly, our security is related to the classical capacity of a malicious party's quantum memory. However, conceptually, the security of the protocol relies on the ability to store quantum information coherently so that a reduction to the quantum capacity or a related quantity would be desirable. This relation has recently been shown for DV protocols using the entanglement cost[20] and the quantum capacity[21,22,40], but its generalization seems challenging for CV protocols as properties of finite groups have been used. Secondly, it is important to derive tight uncertainty relations that hold without additional assumptions. Having such a relation would remove the current constrained on the encoding operation into the quantum memory and possibly also remove the 50% loss limit. Finally, it would be interesting to clarify if OT can be implemented securely in the noisy-storage model using only coherent states. Although squeezing or entanglement is necessary in our security proof, it is not clear whether this is due to our proof technique or whether it is a general requirement.

**Methods**

**Introduction to smooth min-entropy uncertainty relations.** The security of OT in the noisy-storage model relies on tight uncertainty bounds $\lambda^\epsilon$ on the smooth

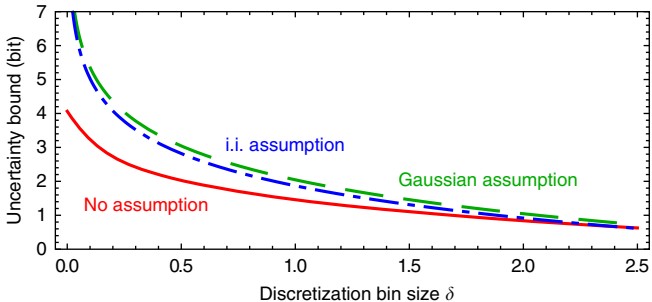

**Fig. 4** Uncertainty bounds. Uncertainty bound without assumptions (red, solid), under the identical and independent (i.i.) assumption over $m_\mathcal{E} = 10$ signals (blue, dashed-dotted) and under Gaussian assumptions (green, long-dashed) depending on the binning size $\delta$. $n = 10^8$, $\epsilon_A = 10^{-7}$. We see that the best bound is obtained under the Gaussian assumption. Moreover, the bound without assumption is very loose for small $\delta$

min-entropy, Eq. (1)[41] (for the details why this is the decisive quantity see method section 4.5). As discussed in the main text we can think of dishonest Bob preparing an ensemble of states $\{\rho_{A^n}^k\}_k$ according to $K$. Here, $A^n$ indicates that Alice ($A$) holds the $n$ modes sent by Bob. A restriction on the encoding map $\mathcal{E}$ translates to a restriction on the ensemble. Clearly, without any restriction on $\mathcal{E}$ there is no restriction on $\rho_{A^n}^k$. If $\mathcal{E}$ is a Gaussian operation, then each $\rho_{A^n}^k$ is a mixture of Gaussian states, since the source distributed by Alice is Gaussian. Note that mixtures have to be considered since combining two or more values of $K$ into one is a simple operation. And finally, if $\mathcal{E}$ acts independently and identically over only $m_\mathcal{E}$ modes, then each $\rho_{A^n}^k$ is identical and independent over $m_\mathcal{E}$ modes since the source is assumed to be identical and independent for each mode.

**Uncertainty relation without assumptions.** Because of the maximization in the definition of the smooth min-entropy over close-by states, it is very difficult to bound it directly. Instead, it is convenient to follow the idea from[42] and to use the fact that it can be related to the conditional $\alpha$-Rényi entropies defined as $H_\alpha(A|B)_\rho = \frac{1}{1-\alpha}\log\mathrm{tr}\left[\rho_{AB}^\alpha(\mathrm{id}_A \otimes \rho_B)^{1-\alpha}\right]$. In particular, it holds for $\alpha \in (1, 2]$ and any two finite random variables $X$ and $Y$ that $H^\epsilon_{\min}(X|Y) \geq H_\alpha(X|Y) - 1/(\alpha-1)\log 2/\epsilon^2$[43]. This relation can be generalized to discrete but infinite random variables using the approximation result from[44]. We then obtain a lower bound on the smooth min-entropy with

$$\lambda^\epsilon(\delta, n) = \sup_{1 < \alpha \leq 2}\left(B^\alpha(\delta, n) - \frac{1}{n(\alpha-1)}\log\frac{2}{\epsilon^2}\right) \quad (3)$$

if $(1/n)H_\alpha(\mathbf{Z}|\theta) \geq B^\alpha(\delta, n)$ holds. Moreover, it suffices to find a bound for $n = 1$, as $B^\alpha(\delta, n) = nB^\alpha(\delta, 1)$[42].

We denote in the following by $\{x_l\}$ and $\{p_l\}$ ($l \in \mathbb{N}$) the probability distribution of the discretized $X$ and $P$ measurement. Using the definition of the $\alpha$-Rényi entropy, one finds that $2^{(1-\alpha)H_\alpha(\mathbf{Z}|\theta)} = \frac{1}{2}\left(\sum_{k \in \mathbb{N}} x_k^\alpha + \sum_{l \in \mathbb{N}} p_l^\alpha\right)$. Since the distributions $x_k$ and $p_k$ are discretized $X$ and $P$ distributions that are related by Fourier transform, they satisfy certain constraints. For instance, it is not possible that both have support only on a finite interval.

A precise formulation of the constraint for the probabilities $x[I]$ and $p[J]$ to measure $X$ in interval $I$ and $P$ in interval $J$ has been given by Landau and Pollak[45]. They proved that these probabilities are constrained by the inequality $\cos^{-1}\sqrt{q[I]} + \cos^{-1}\sqrt{p[J]} \geq \cos^{-1}\sqrt{\gamma(|I|, |J|)}$. Here, $|I|$ denotes the length of the interval $I$, and $\gamma(a, b) := ab/(2\pi\hbar)S_0^{(1)}\left(1, ab/(4\hbar)\right)^2$ with $S_0^{(1)}$ the 0th radial prolate spheroidal wave function of the first kind. For $ab$ sufficiently small $\gamma$ can be approximated by $\gamma(a, b) \approx ab/(2\pi\hbar)$.

The above constraint on $q[I]$ and $p[J]$ can be reformulated in the following way[46]: (i) if $0 \leq q[I] \leq \gamma(|I|, |J|)$, then all values for $p[J]$ are possible, and (ii) if $\gamma(|I|, |J|) \leq q[I]$, then $p[J] \leq g(q[I], |I|, |J|)$ for $g(q, a, b) := \left[\sqrt{q\gamma(a, b)} + \sqrt{(1-q)(1-\gamma(a, b))}\right]^2$. This yields an infinite number of constraints for $\{q_l\}$ and $\{p_l\}$. Let us assume that $\{q_l\}$ and $\{p_l\}$ are decreasingly ordered, then for all $M, N \in \mathbb{N}$ it has to hold that

$$\sum_{j=1}^N p_j \leq g\left(\sum_{i=1}^M q_i, M\delta, N\delta\right). \quad (4)$$

It is challenging to turn the above constraints into an explicit and tight upper bound for the $\alpha$-Rényi entropy. In the following we discuss a possible way that connects the above constraints with a majorization approach.

Let us denote by $\{r_j\}$ the decreasingly ordered joint sequence of both distributions $\{q_l\}$ and $\{p_l\}$. Then, we can write $2^{(1-\alpha)H_\alpha(\mathbf{X}|\theta)} = \frac{1}{2}\sum_{j \in \mathbb{N}} r_j^\alpha$. Since the function $r \mapsto \sum_{j \in \mathbb{N}} r_j^\alpha$ is Schur convex, it can be upper bounded by any sequence $w_j$ majorizing $r_j$. Such a $w_j$ can be constructed in a way shown in ref.[47].

First, note that condition (ii) above implies that $q[I] + p[J] \leq q[I] + g(q[I], |I|, |J|)$. Optimizing the right hand side over all $0 \leq q[I] \leq 1$, we obtain the constraint $q[I] + p[J] \leq 1 + \sqrt{\gamma(|I|, |J|)}$. This then implies that $\sum_{j=1}^n r_j \leq 1 + F_n(\delta)$, where $F_n(\delta) = \max_{1 \leq k \leq n}\sqrt{\gamma(k\delta, (n-k)\delta)}$. Here, the maximum is attained for $k = \lfloor\frac{n}{2}\rfloor$.

We can now construct a majorizing sequence $w$ by setting recursively $w_1 = 1$ and $w_k = F_k - w_{k-1}$ for $k \geq 2$. The obtained bound on the $\alpha$-Rényi-entropy is then given by $B^\alpha_{\mathrm{Maj}} = \frac{1}{1-\alpha}\log\left(\frac{1}{2}\sum_k w_k^\alpha\right)$. According to Eq. (3), this translates into a bound on the smooth min-entropy given by

$$\lambda^\epsilon_{\mathrm{Maj}} := \sup_{1 < \alpha \leq 2}\left(B^\alpha_{\mathrm{Maj}} - \frac{1}{n(\alpha-1)}\log\frac{2}{\epsilon^2}\right). \quad (5)$$

A plot of the bound is given in Fig. 4. We emphasize that the obtained bound seems not very tight, especially for small $\delta$. We believe that this problem is due to the fact that the way how the majorizing sequence is constructed does not exploit all the possible constraints.

**Uncertainty relation under Gaussian assumption.** In order to obtain an improved uncertainty relation we assume that the states $\rho_{A^n}^k$ are mixtures of Gaussian states. Similarly as before, we derive a bound for the $\alpha$-Rényi entropy with $\alpha \in (1, 2]$ and use Eq. (3) to obtain a bound on the smooth min-entropy. This argument implies that it is again sufficient to consider the case $n = 1$.

Let us first assume that the state is Gaussian such that the continuous probability distributions $x(s)$ and $p(s)$ of the $X$ and $P$ measurements are Gaussian. We denote the standard deviations of the $X$ and $P$ distribution by $\sigma_X$ and $\sigma_{P,\alpha}$ respectively. Using Jensen's inequality we can upper bound $x_k^\alpha = \left(\int_{I_k} x(s)ds\right)^\alpha \leq \delta^{\alpha-1}\int_{I_k} x(s)^\alpha ds$, where $I_k$ denotes the interval corresponding to the bin $k$. Taking now the sum over all bins we arrive at $\sum_k x_k^\alpha = \delta^{\alpha-1}\int x(s)^\alpha ds =: g(\tilde\sigma_X)$, where $g(x) = 1/\left[\sqrt{\alpha}\left(\sqrt{2\pi}x\right)^{\alpha-1}\right]$ and $\tilde\sigma_X = \sigma_X/\delta$ is the relative standard deviation of the Gaussian distribution $x(s)$.

Note that the bound $g(\tilde\sigma_X)$ becomes very loose if $\tilde\sigma_X$ is very small and can even become larger than the trivial upper bound 1. We avoid that problem by simply bounding $\sum_k q_k^\alpha \leq \min\{g(\tilde\sigma_X), 1\}$. The same applies to the $P$ quadrature yielding the upper bound $\sum_k q_k^\alpha + \sum_l p_l^\alpha \leq \min\{g(\tilde\sigma_X), 1\} + \min\{g(\tilde\sigma_P), 1\}$. We can now apply Kennard's uncertainty relation for the standard deviations of $X$ and $P$ to obtain $\tilde\sigma_X\tilde\sigma_P \geq \hbar/(2\delta_x\delta_p)$[48]. Optimizing $\min\{g(\tilde\sigma_X), 1\} + \min\{g(\tilde\sigma_P), 1\}$ over all possible $\tilde\sigma_X, \tilde\sigma_P$ gives $1 + (\delta^2/(\pi\hbar))^{(\alpha-1)}/\alpha$. Hence, we find for Gaussian states the uncertainty relation $H_\alpha(\mathbf{Z}|\theta) \geq B^\alpha_{\mathrm{Gauss}}(\delta, \mathbf{n})$ with

$$B^\alpha_{\mathrm{Gauss}}(\delta, n) := \frac{n}{1-\alpha}\log\frac{1}{2}\left(1 + \frac{1}{\alpha}\left(\frac{\delta_x\delta_x}{\pi\hbar}\right)^{(\alpha-1)}\right). \quad (6)$$

This relation then leads to a bound on the smooth min-entropy with $\lambda^\epsilon_{\mathrm{Gauss}}(n)$ via Eq. (3). The improvement over the previous bound can be seen in Fig. 4.

Let us finally show that this relation also holds for arbitrary mixtures of Gaussian states. Let us take $\rho = \sum_y \mu_y\rho^y$ with probability $\mu_y$ and $\rho^y$ a Gaussian state for any $y$. We then obtain that $\sum_k x_k^\alpha = \sum_k\left(\sum_y \mu_y\int_{I_k} x^y(s)ds\right)^\alpha \leq \sum_k \sum_y \mu_y\left(\int_{I_k} x^y(s)ds\right)^\alpha = \sum_y \mu_y\sum_k\left(x_k^y\right)^\alpha$. Here we denote by $x^y$ the $X$ probability distribution of $\rho^y$, and we used the concavity of the function $x \mapsto x^\alpha$. This argument shows that the above uncertainty relation extends to arbitrary (even continuous) mixtures of Gaussian states.

**Uncertainty relation under the identical and independent assumption.** Let us assume that a certain number of quantum states are identical and independent, i.e. that each $\rho_{A^n}^k$ has tensor product structure $\rho_{A^n}^k = \left(\sigma_{A^{m_\mathcal{E}}}^k\right)^{\otimes n/m_\mathcal{E}}$, with $n/m_\mathcal{E}$ being an integer. It is known that if $n/m_\mathcal{E}$ goes to infinity, the smooth min-entropy converges to the Shannon entropy[49,50]. More precisely, we can lower bound $\frac{1}{n}H^\epsilon_{\min}(\mathbf{Z}^n|\theta^n)$ by

$$\frac{1}{m_\mathcal{E}}H(\mathbf{Z}^{m_\mathcal{E}}|\theta^{m_\mathcal{E}}) - 4\sqrt{\frac{m_\mathcal{E}}{n}}\log(\Gamma(\mathbf{Z}^{m_\mathcal{E}}))^2\sqrt{\log\frac{2}{\epsilon^2}}, \quad (7)$$

where $\Gamma(\mathbf{Z}^{m_\mathcal{E}}) = 2 + 2^{H_{1/2}(\mathbf{Z}^{m_\mathcal{E}})}$. This relation has also been shown for infinite-dimensional alphabets in[44]. If we assume that Alice knows the covariance matrix of her reduced state, we can bound $H_{1/2}(\mathbf{Z}^{m_\mathcal{E}})$, and thus, $\Gamma(\mathbf{Z}^{m_\mathcal{E}})$. It therefore remains to find a lower bound on the Shannon entropy $H(\mathbf{Z}^{m_\mathcal{E}}|\theta^{m_\mathcal{E}})$.

For simplicity let us assume that $m_\mathcal{E} = 1$. Because the measurement choice $\theta$ is uniformly distributed, we find that $H(Z|\theta) = 1/2(H(X_\delta) + H(P_\delta))$. Thus, we recover the usual entropic uncertainty relation for the Shannon entropy which has been extensively studied. In particular, it has been shown that $H(X_\delta) + H(P_\delta) \geq \log(e\pi\hbar/\delta^2)$[51]. It is easy to see that in the case of an arbitrary $m_\mathcal{E}$, we similarly obtain $H(\mathbf{Z}^{m_\mathcal{E}}|\theta^{m_\mathcal{E}}) \geq m_\mathcal{E}/2\log(e\pi\hbar/\delta^2)$. In conclusion, we arrive at an uncertainty

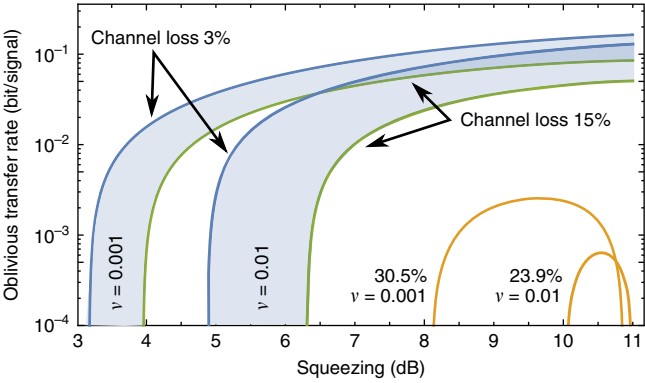

**Fig. 5** Simulation of oblivious transfer rate. Simulation of oblivious transfer rate under Gaussian assumption versus the amount of squeezing used to prepare the EPR state for two different storage rates $\nu = 0.01$ and $\nu = 0.001$. The shaded areas correspond to a channel loss between 3 and 15% which is the experimentally investigated region. The orange traces are calculated for a channel loss close to the maximum possible value for the respective storage rate. The anti-squeezing of the employed squeezed state was calculated using parameters characterizing the experimental squeezed light sources and homodyne detectors: 98.2% escape efficiency, 79.8 mm optical round-trip length, 8 MHz sideband frequency, 91.4% total optical efficiency. Other parameters: information reconciliation efficiency 92.5%, $n = 2 \times 10^5$ samples, $\epsilon_A = 10^{-7}$, $\alpha_{cut} = 51.2$, $\delta = 0.1$, $N_{max} = 100$, transmittance of quantum memories 0.75

relation with

$$\lambda_{IID}^\epsilon(\delta, n) = \frac{1}{2}\log\left(e\pi\hbar/(\delta_x\delta_p)\right) - 4\sqrt{\frac{m_\mathcal{E}}{n}}\log(\Gamma(\mathbf{Z}^{m_\mathcal{E}})^2)\sqrt{\log\frac{2}{\epsilon^2}}. \quad (8)$$

**Security proof against a malicious Bob with restricted memory.** The security proof for an honest Alice is similar to the one in ref. [28], which is using key results from ref. [17,19]. The main difference is that we have to include the information reconciliation leakage, and to take into account that Bob's quantum memory can be infinite-dimensional and that $K$ can be continuous. According to the protocol, we can assume that Alice is distributing a state $\rho_{AB}$, $A = A_1, ..., A_n$, for which

$$\text{tr}\left(\rho_{A_i}(q[-\alpha_{cut}, \alpha_{cut}])\right) \geq p_{\alpha_{cut}}, \quad (9)$$

$$\text{tr}\left(\rho_{A_i}(p[-\alpha_{cut}, \alpha_{cut}])\right) \geq p_{\alpha_{cut}} \quad (10)$$

holds for any mode $i$. As in the main text $Z$ denotes Alice's discretized measurement outcomes with the binning $(-\infty, -\alpha_{cut} + \delta]$, $(-\alpha_{cut} + \delta, -\alpha_{cut} + 2\delta]$, ..., $(\alpha_{cut} - 2\delta, \alpha_{cut} - \delta]$, $(\alpha_{cut} - \delta, \infty)$. Note, that $\alpha_{cut}$ is an integer multiple of $\delta$. We further introduce $\tilde{Z}$ as the string of outcomes if Alice would measure a uniform binning of $\delta$ over the entire range $\mathbb{R}$ (as used in the derivation of the uncertainty relations).

To ensure composable security for Alice, we have to show that for any memory attack of Bob, there exists a random variable $D$ in $\{0, 1\}$ such that conditioned on $D = d$, Bob does not know $\mathbf{s}_d$ with probability larger than $1 - \epsilon_A$[19]. Denoting by $\mathbf{B}'$ all the classical and quantum information held by a malicious Bob at the end of the protocol, this condition can be formulated by using the trace norm

$$\left\| \rho_{S_D S_{\overline{D}} D\mathbf{B}'} - \tau_{S_D} \otimes \rho_{S_{(1-D)}D\mathbf{B}'} \right\|_1 \leq \epsilon_A, \quad (11)$$

where $\tau_{S_D}$ denotes the uniform distribution over $S_D$. We use lower indices to indicate the relevant systems, that is, the overall state of a joint system with quantum information $A$, $B$ and classical random variables $X$, $Y$ is denoted by $\rho_{ABXY}$. Hence, if $X$ is a random variable, $\rho_X$ denotes its distribution, if $A$ is a quantum system, $\rho_A$ denotes its quantum state, and its combination $\rho_{XA}$ can conveniently be described by a classical quantum state.

Recall that $\mathbf{s}_0, \mathbf{s}_1$ are obtained by hashing the substrings $\mathbf{Z}_0, \mathbf{Z}_1$. Choosing the length $\ell$ of the bit strings $\mathbf{s}_0, \mathbf{s}_1$ sufficiently small has the effect of randomization and destruction of correlation, i.e., establishing Eq. (11). More precisely, the

condition from Eq. (11) is satisfied if[52]

$$\ell \leq H_{min}^{\epsilon_1}(\mathbf{Z}_D|S_{\overline{D}}D\mathbf{B}) - 2\log_2\frac{1}{\epsilon_A - 4\epsilon_1} \quad (12)$$

and we can optimize over $0 < 4\epsilon_1 < \epsilon_A$. The crucial difference to the discrete-variable case is that the above relation holds even if Bob's quantum memory is modeled by an infinite-dimensional system.

Bob's information $\mathbf{B}'$ consist of the states of his quantum memories $\mathbf{Q}$ and his classical register $\mathbf{K}$ (see Fig. 1), Alice's basis choice $\boldsymbol{\theta}_A$, and the information reconciliation syndrome $\mathbf{W} = (\mathbf{W}_0, \mathbf{W}_1)$. The next goal is to remove the conditioning on the quantum system by relating it to the classical capacity of Bob's quantum memory $\mathcal{F}_{\Delta t}^{\otimes \nu n}$. For this step we use the key result from ref. [19] which says that $H_{min}^{\epsilon_1}(\mathbf{Z}_D|S_{\overline{D}}D\mathbf{Q}\mathbf{K}\boldsymbol{\theta}_A\mathbf{W})$ is larger than minus the binary logarithm of

$$\mathcal{P}_{succ}^{\mathcal{F}_{\Delta t}^{\otimes \nu n}}\left(\left\lfloor H_{min}^{\epsilon_2}(\mathbf{Z}_D|D\mathbf{K}\boldsymbol{\theta}_A\mathbf{W}) - \log_2\frac{1}{(\epsilon_1 - \epsilon_2)^2}\right\rfloor\right), \quad (13)$$

where $\mathcal{P}_{succ}^{\mathcal{F}_{\Delta t}^{\otimes \nu n}}(\ell)$ is the success probability to send $\ell$ bits through the channel $\mathcal{F}_{\Delta t}^{\otimes \nu n}$. Again, we have the freedom to optimize over all $0 < \epsilon_2 < \epsilon_1$. The above result, originally proven for finite dimensions, can easily be extended to infinite-dimensions using the finite-dimensional approximation results from ref. [44]. For the following, we will assume that the reliable transmission of classical information over the channel $\mathcal{F}_{\Delta t}$ decays exponentially above the classical capacity, i.e., $\mathcal{P}_{succ}^{\mathcal{F}_{\Delta t}^{\otimes n}}(nR) \leq 2^{-\xi(R - \overline{C}_{cl}(\mathcal{F}_{\Delta t}))}$, usually referred to as a strong converse.

The final step is to lower bound the smooth min-entropy of $\mathbf{Z}_D$ given $D\mathbf{K}\boldsymbol{\theta}_A\mathbf{W}$ in Eq. (13). It is convenient to use the min-entropy splitting theorem[14] saying that for given two random variables $\mathbf{Z}_0, \mathbf{Z}_1$, there exists a binary variable $D$ such that the smooth min-entropy of $\mathbf{Z}_D$ given $D$ is larger than half of the smooth min-entropy of the two strings together, that is, $H_{min}^\epsilon(\mathbf{Z}_D|D\boldsymbol{\theta}_A\mathbf{W}) \geq \frac{1}{2}H_{min}^\epsilon(\mathbf{Z}_0\mathbf{Z}_1|\boldsymbol{\theta}_A\mathbf{W}) - 1$. This theorem defines retrospectively the random variable $D$. The conditioning on the information reconciliation syndrome $\mathbf{W}$ can be removed by simply subtracting the maximum information contained in $\mathbf{W}$ given by $nr_{IR} = \log_2|\mathbf{W}|$. Before we can apply the uncertainty relation, we have to eventually relate the entropy of $\mathbf{Z}$ by the one of $\tilde{\mathbf{Z}}$. This is necessary since a state-independent uncertainty relation cannot be satisfied for quadrature measurements with a finite range. But due to the condition that $\alpha_{cut}$ is chosen such that the probability to measure an event outside the measurement range is small, we can bound[53] the $\epsilon_2$-smooth entropy of $\mathbf{Z}$ given by the $(\epsilon_2 - \epsilon_{\alpha_{cut}})$-smooth entropy of $\tilde{\mathbf{Z}}$, where $\epsilon_{\alpha_{cut}} = \sqrt{2\left(1 - p_{\alpha_{cut}}^n\right)}$. Note that this step requires that $\epsilon_{\alpha_{cut}} < \epsilon_2 < \epsilon_A/4$. Since the probability that Alice measures an outcome with absolute value larger than $\alpha_{cut}$ only depends on her reduced state, the same holds conditioning on $\mathbf{K}$ and $\theta_A$[53].

Hence, given that the uncertainty relation from Eq. (1) holds, we find that $\epsilon_A$-security for Alice as in Eq. (11) is satisfied, if we choose

$$\ell = \frac{n}{2}\xi(r_{OT} - \nu\mathcal{C}_{cl}(\mathcal{F})) - \log\frac{1}{\epsilon_A - 4\epsilon_1}, \quad (14)$$

where

$$r_{OT} := \frac{1}{2}\left(\lambda^{\epsilon_2 - \epsilon_{\alpha_{cut}}}(n) - r_{IR} - \frac{2}{n}\left(\log\frac{1}{(\epsilon_1 - \epsilon_2)^2} + 1\right)\right). \quad (15)$$

The length $\ell$ can be optimized over all $\epsilon_1, \epsilon_2 \geq 0$ arbitrary such that $\epsilon_A > 4\epsilon_1 > 4\epsilon_2 > 4\epsilon_{\alpha_{cut}}$. We then obtain Eq. (2) in the main text for a Gaussian loss channel satisfying $\xi = 1$[32].

Figure 5 shows a simulation of the oblivious transfer rate under Gaussian assumption versus the amount of squeezing in the EPR state. For the experimentally investigated region of channel loss the generated EPR state was close to optimal. Only for loss very close to the maximum channel loss the optimal squeezing value is around 10 dB.

**Experimental parameters**. The squeezed light sources were pumped with 140 and 170 mW, respectively. The local oscillator power for Alice's and Bob's homodyne detector was 10 mW each yielding a vacuum-to-electronic-noise clearance of about 18 dB. The 14 bit analog-to-digital converter allowed us to measure a maximum $\alpha$ of about 100. The quantum efficiency of the photo diodes was 99%, the homodyne visibility 98%. The phases of the local oscillators were randomly switched at a rate of 100 kHz between the amplitude and phase quadrature using a fiber coupled waveguide phase modulator. The reconstructed covariance matrix measured without loss in the channel and after local rescaling of Bob's variances reads

$$\begin{pmatrix} 21.93 & (0) & -21.84 & (0) \\ (0) & 24.89 & (0) & 24.80 \\ -21.84 & (0) & 21.93 & (0) \\ (0) & 24.80 & (0) & 24.89 \end{pmatrix}, \quad (16)$$

where the entries in brackets were not measured, but assumed to be 0. Taking an upper bound on the variance of Alice's state of 25, $\epsilon_A = 10^{-7}$ and $n = 2 \cdot 10^5$ we obtain a minimum $\alpha_{cut} \approx 47.9$ using the expression in the previous section.

For further experimental details we refer to ref. [24].

For the post-processing we used C++11 as programming language, compiled with GNU GCC 6.3, and ran the binary on a single core of an Intel Xeon E7-8870v2 CPU in a PC running Linux (Debian 8) as operating system. On average we achieved a rate of approximately 1k oblivious bit transfers per second.

**Data availability**. Codes for calculating the oblivious transfer rate are available at https://github.com/qpit/ObliviousTransfer. All other data are available from the authors upon request.

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

## Acknowledgements

We would like to thank Anthony Leverrier, Loïck Magnin and Frédéric Grosshans for useful discussions about the continuous-variable world. F.F. is supported by the Japan Society for the Promotion of Science (JSPS) by KAKENHI grant No. 24-02793. T.G. is supported by the Danish Council for Independent Research (Individual Postdoc and Sapere Aude 4184-00338B). C.S. is supported by a NWO VIDI grant. S.W. is supported by STW Netherlands, as well as an NWO VIDI and an ERC Starting Grant. The experimental work is partially supported by the Deutsche Forschungsgemeinschaft (project SCHN 757/5–1).

## Author contributions

F.F. and S.W. conceived the project. F.F., C.S., and S.W. developed the security proof, T.G. and R.S. performed the experimental implementation, C.P. implemented the

information reconciliation and classical post-processing, and F.F. did the numerical simulations. F.F., T.G., and C.P. wrote the manuscript with contributions from all authors.

## Additional information

**Competing interests:** The authors declare no competing interests.

