## [Peer Review File(PDF 459 kb) · Nature Communications]

Reviewers' comments:

Reviewer #1 (Remarks to the Author):

In this work the authors present a continuous variable (CV) oblivious transfer (OT) protocol, a two-party cryptographic primitive between distrustful parties that can be used as a building block for bit commitment and secure identification. On the theory side, the authors apply recent advances in infinite-dimensional cryptographic tools to derive new entropic uncertainty relations appropriate for a mutually distrustful Bob and Alice and extend the noisy-storage model analysis to the CV regime for the first time. The authors also carry out an experimental demonstration using a highly entangled EPR resource, including complete end-to-end data processing (error reconciliation and privacy amplification) which can be very challenging in the CV context. For low loss channels the secure OT under a Gaussian encoding assumption rate was found to be 0.1 - 0.05 bits per measured signal which is around an order of magnitude higher than the previous DV demonstration.

Overall, these are interesting results and the manuscript itself is generally of high quality. There are a few key changes and additions that need to be made to improve the clarity and reproducibility of the results and several minor comments and corrections, but I suspect these will all be able to be successfully addressed. However, overall I'm not sure about how novel and influential the presented results are. Firstly, it seems to me that the theoretical extensions to the CV case are impressive without being groundbreaking. Given the existence of previous technical breakthroughs in Ref's 44 and 53 (by some of the same authors as here) which extended the composable security framework, conditional smooth min-entropies, and the derivation of several entropic uncertainty relations to the infinite-dimensional case, it is a valuable, but relatively straightforward, task to derive an CV version of an uncertainty relation in the style of Ref. 42 which in combination with the strong converse of Ref. 32 allows one to follow the analysis of Refs. 20 and 28.

That might be alright if this then allowed an experimental demonstration of noticeably greater performance, but overall it's not clear that this is the case. I should emphasise that this is not due to a poor quality experimental implementation (to the contrary, the EPR resource used here is, to my knowledge, virtually the best in the world) but rather due to in-principal limitations. However, this leads the reader to conclude that, with these current techniques, unclear if there is much benefit in carrying out CV-OT protocols. A more detailed version of these concerns follows below, along with some important substantive questions/criticisms and finally some minor comments.

%% Comparative weakness in performance %%

a) Strength of assumptions on the memory. To get the best results that they report, the authors need to introduce additional assumptions regarding the encoding operation into the noisy memory. They consider both the case where Bob can only act in an iid fashion on blocks of a fixed length and also the case of Gaussian encoding, with the latter assumption key for achieving the best results. The authors are very up-front about this and the issue is well explained. However, whilst the iid assumption seems conceptually similar to noisy storage itself (essentially a limit on the “depth” of encoding operations) the Gaussian assumption seems a lot more restrictive given that there are already experimental demonstrations of reasonably high-quality non-Gaussian operations. A lot of these are non-deterministic but in this paper the authors consider a storage rate ν (which would include non-deterministic encoding operations) between .1 and 1% so there are potentially already encoding operations available today that would violate the Gaussian assumption. Of course it’s not at all clear that non-Gaussian encodings would actually prove beneficial in the end, but until the question is settled one way or the other this seems to be a non-trivial extra assumption in comparison to DV-OT.

b) The 50% loss limit. This also seems like a significant comparative weakness, although there isn’t actually a clear comparison. Starting on line 360 the authors explain that to achieve any secure OT the channel transmission between Alice and Bob must be above 50% even if all other aspects of the implementation are perfect. From my reading of Ref. 24 the same restriction does not hold there. That said, there is an overall tradeoff in Ref.24 between the transmission and the error rate, but my understanding is that at least for an otherwise idealised implementation the transmittance can be arbitrarily low. On the other hand, for many OT applications such as identification I suppose there are operational scenarios such as ATM’s where short distance, low loss regime are the relevant ones? Or perhaps for practical non-idealised settings CV-OT is still at a comparative advantage? I would point out that, whilst the comparison to [24] itself shows higher rates, Ref. 24 itself was a quite high loss implementation. For instance in a state-of-the-art table top experiment today could have much higher detection efficiencies than 48% that appear there. In any event, the authors should explain the relative CV-DV loss tolerance in more detail.

c) Possibly, the status of the strong converse theorem that is utilised (see comment e) but I think this is actually not a problem.

%% Major comments %%

d) Insufficient details about the experiment to allow for reproduction of results. In particular the observed covariance matrix is not reported anywhere in the paper, which makes it impossible to attempt to reproduce the curves given in Fig. 3 b). The authors reference [16] but I couldn’t find a CM there instead [16] directs to [27]. This is already more work than the reader should have to do to check the results. Also, I assume that the experiment was literally carried out using the same data set reported in [27]. This lack of experimental data also makes it impossible to verify other crucial

parts of the security proof such as the maximum codeword photon number (comment e) and the necessary detector range (see f).

e) The wording regarding the strong converse for lossy bosonic channels is ambiguous at times. As explained in the paper, to get decent rates a strong converse is necessary to ensure exponential failure of the memory when trying to store information above the classical rate. Also it is explained that Ref. 32 has shown a strong converse for a lossy bosonic channel, provided all of the codewords are contained with high probability in a finite photon-number subspace. However in line 289 and later in 658 the authors write “we will assume that the reliable transmission of classical information over the channel decays exponentially above the classical capacity”. But, if I understand the discussion beginning in line 331, then this assumption is in fact definitely true for this protocol because it satisfies the requirements of Ref 32. Specifically, for the source used although the codeword states do not themselves have a definite photon number, their support lies almost completely within some photon subspace with $n < N_{\max}$. The authors should state this explicitly and (as mentioned above) quote the measured value of N_{\max} determined from the source calibration.

f) The choice of the α_{cut} parameter. As explained in the paper, any given security parameter ϵ_A fixes a strict lower bound for α_{cut} , i.e. it must be the case that the detectors used are sensitive out to that value and that the source is calibrated such that any value outside that range only occurs with a probability $\epsilon_{\{\alpha_{\text{cut}}\}}$. Currently, the paper provide neither the assumptions nor the experimental data to used to find this lower bound. By rearranging the formula's given on page 9. I calculate the expression for the minimum allowable value for the detection probability to be $p_{\alpha_{\text{cut}}} > 1 - \epsilon_A^2/32$. If one assumes the quadrature measurements are well approximated by a Gaussian of variance V then one has $p_{\{\alpha_{\text{cut}}\}} = \text{erf}(\alpha_{\text{cut}}/\sqrt{2V})$ which gives $\alpha_{\text{cut}} > \sqrt{2V} \text{erfinverse}[(1 - \epsilon_A^2/32)^{1/n}]$. Taking the worst case value from [27] of $V = 26$ if calculate the lower bounds to be 48 for $n = 10^5$ and 51.8 for $n=10^8$. So it seems that the value used in this paper is okay as long as the data is indeed similar to [27]. But the authors should make this calculation more explicit and, as above, provide variances to allow the reader to check this condition on α_{cut} is satisfied.

g) It would be good if there could be some commentary on the optimality or otherwise of the experimental implementation. Specifically, is it actually a practically better strategy to mix two squeezed states to make an EPR state as opposed to a prepare and measure protocol where Alice randomly sends either phase or amplitude squeezed states? And are the squeezing values used in this experiment optimal? In CVQKD situations it is often optimal to use low modulation variance (or low squeezing) in high loss scenarios.

h) In the initial discussion the authors characterise the memory in terms of the time Δt that Alice makes Bob wait. But in the actual calculations it is characterised simply in terms of the effective

loss of the channel. There doesn't seem to be any discussion of roughly how long Δt should be in order to correspond to a certain loss in realistic quantum memories.

Minor comments

i) Above, I listed information that must be included in order to meet basic standards of reproducibility. In addition, given that it is usually a quite lengthy process to actually calculate rates from expressions like Eq. (2) it would be great to see the codes that were used to evaluate the expressions included as supplementary material. I wouldn't insist on this but for purposes of transparency and reproducibility I'm sure it would be a very valuable addition.

j) Line 392 the Section number is missing

k) line 401 "bit per symbol" should be "bits per symbol".

l) line 419 "taken experimental data" should read "experimental data taken"

m) line 591 "lose" should be "loose"

n) line 614, its probably a bad idea to use η symbol again given it is already representing loss.

Reviewer #2 (Remarks to the Author):

In the submission, an experimental demonstration of a quantum cryptographic protocol, namely 1-out-of-2 oblivious transfer (OT), is performed under the noisy storage model for continuous variable.

While I find the work scientifically interesting, I have some questions/comments about it. Therefore, I will withhold my judgement until I see a revised version of the paper. My detailed comments are as follows:

Main Comments:

1) Motivation:

As noted in the Introduction of the paper, the protocol oblivious transfer has already been demonstrated in previous experiments in Refs. [19,24] for the case of discrete variable. In view of these demonstrations, why would yet another demonstration be of broad interest and scientifically important?

Therefore, I suggest that the authors to provide a clear discussion about the motivation of their work. Why is it interesting and important to employ continuous variable for the specific application of 1-out-of-2 OT under the noisy storage model?

2) Differentiating between distance and loss:

In the second paragraph of column 2 of p. 6, it states that "For short distances (i.e., low channel losses)," . Here, the statement seems to equate channel losses with distances. In my opinion, such a statement is confusing because it completely ignores coupling losses in optical components that are a fact of life and are inevitable in any fiber-based implementation such as the one in Ref. [24]. In fact, a single fiber-optics component such as a coupler could easily give a 1dB or so loss that would render a major modification to the secure OT rate in a continuous variable implementation like the submission.

In summary, my view is that CV implementation of 1-out-of-two OT is only feasible in a low-loss regime. And, such a low loss regime is only realistic in a very short-distance open air setting where various focusing techniques including lenses could be used to increase in the collection efficiency. However, CV implementation of 1-out-of-two OT is unrealistic in a fiber-based setting because even a single fiber-optics component could easily produce a big loss (say 1dB) to the channel that would substantially lower the rate of secure 1-out-of-two OT. Perhaps, these limitations need to be discussed in the paper.

3) Fair Comparison with previous results:

It is stated in the submission that “For short distances (i.e., low channel losses), we obtain rates that are about a magnitude larger than those achieved in the DV case [24] while using significantly smaller block sizes of about 10^5 compared to 10^7 .”

I note that the DV implementation [24] was done with fiber-optics components which introduce substantial losses (e.g. order 1dB loss per component). The success of Ref. [24] under such high loss demonstrates the robustness of DV implementation, compared to CV implementation (for the case of 1-out-of-2 oblivious transfer (OT) under the noisy storage model). With such a high loss, CV implementation in the current submission would probably have given a zero rate.

Another point to note is that it is well known that while information reconciliation can be efficiently done for DV QKD, the construction of an efficient method for information reconciliation that takes finite size effects into account remains a major challenge for many CV QKD protocols. For this reason, I am not sure that a comparison on block size only completely captures the demanding situation of CV implementations.

Other comments:

4) In the last line of the second column of p. 1, it states that “We overcome this problem by assuming that the dishonest party’s storage operation is Gaussian.”

This seems like a rather strong assumption. I wonder if the authors could provide some justification to such an assumption or discuss about its limitations.

5) In line 205, I wonder if the authors could state clearly what value of ϵ_C they are trying to achieve in the present submission and why.

6) In line 353-356 “The weakest requirements on the parameters have to be imposed under the Gaussian assumption in which security can already be obtained for low numbers of signals n approximately 10^5 .” Perhaps, it will be useful to provide a reference here.

Also, is such a conclusion only valid under the Gaussian assumption? Please discuss about the cases of no assumption and i.i.d. assumption.

7) In line 360-362, it states that “In general, to obtain security a transmittance of the channel between Alice and Bob larger than 0.5 and non-trivial squeezing is required.”

Perhaps, it would be useful for the authors to discuss in practice how hard it is to achieve such a low loss in experimental implementation. Also, explore how much squeezing is required and how hard it is to achieve such an amount of squeezing in a practical setting.

8) Starting from line 367, an experimental implementation is presented. However, I find the description lacking as a number of important details such as wavelength and distance are missing.

9) In lines 384-386, it states that “Optical loss in this channel was simulated by a variable beam splitter comprising a half-wave plate and a polarizing beam splitter.”

I don't understand this statement. Wasn't it a real experiment that your team has performed? How do you do the simulation here?

10) In lines 396-399, it states that “We then chose from each set the first 10^5 for post-processing (i.e., $n = 2 \times 10^5$) to keep the block size of the information reconciliation code constant.”

Doesn't this violate the fair sampling assumption? Why is the protocol secure?

11) On line 418, it states that $\epsilon_{IR} = 10^{-3}$. This seems quite large (compared to standard QKD security parameters of about 10^{-10}). Is there any reason for such a large number?

12) On line 430, it says “the lossless channel”. Why consider only a lossless channel?

13) On line 430, it says that “We see that for short distances...” As mentioned in one of the Main Comments above, I think it is important to differentiate clearly between short distances and losses and state clearly if one is considering free-space or fiber-based setting.

14) On line 435, are the numbers “ $\nu=0.01$ ” and “ $\nu=0.001$ ” given there realistic? Why?

15) On lines 439-440, it states that “showed security against a malicious party with an imperfect quantum storage device.” Perhaps, you mean “highly imperfect” quantum storage devices because there is severe limit on the rate of OT due to the imperfections of quantum storage devices.

16) On lines 444-446, “However, security can also been obtained with weaker entangled sources than used in our experiment such that on chip entanglement sources are possible [39].” If one includes coupling loss or collection efficiency loss into consideration, I am not sure if the above statement is still correct. Any thought?

Reviewer #3 (Remarks to the Author):

The paper presents an experimental implementation of quantum oblivious transfer using optical beams, continuous variables. Oblivious transfer is a secure quantum communication protocol in which a piece of information is transferred from sender to receiver such that the sender remains oblivious to which exactly information has been transferred. Oblivious transfer is a fundamental primitive in cryptography. While perfect information theoretic security is impossible, quantum oblivious transfer protocols can limit the dishonest players' cheating. It is well known that information theoretic security is considered impossible for oblivious transfer in both the classical and the quantum world. Noisy-storage model is used in this particular implementation to enable the secure implementation of the oblivious transfer. It assumes that the quantum memory device of a malicious party is imperfect (noisy) and is used in a number of two-party cryptographic primitives where unconditional security cannot be reached.

The paper is well written and the results are scientifically sound. It is a very interesting and promising approach as a continuous-variable (CV) platform is highly compatible with real world telecommunication systems and relatively easy to handle. Also the sending rates can be quite high, which together with high quantum efficiency of homodyne detection used makes the overall system very effective. The challenging part is the security proof; the approaches from the discrete variables (DV) protocols cannot be easily translated over. The work is quite unique as oblivious transfer is one of the most important cryptographic primitives and this is I believe the first experimental implementation of such types of two-party protocols in CV regime. Also in DV regime, the only experimental implementation, also using noisy-storage model, has been performed quite recently, in 2014, and a bit earlier a bit-commitment protocol, in the same model. This current paper presents a neat experimental work and an advanced novel security proof, of importance both for the fundamental understanding of secure quantum protocols and for applications in quantum

technologies. I fully recommend the publication in Nature Communications. I have a few comments though.

The security proof is based on new entropic uncertainty relation for the field quadratures, or more precisely on founding a good bound for the smooth entropy. The equation (1) is an important result, not only for the OT protocol, many CV secure protocols rely on properly bounding smooth-min entropy for CV platform, which is still largely an open question. It would be very helpful to have a comparison with the state-of-art and a comment on applicability of the derived inequality beyond this particular protocol.

I do not quite understand why it is more convenient to use the two-mode squeezed vacuum rather than doing a prepare-and-measure scheme. Would switching to prepare-and-measure scheme relax the loss limit? The EPR states are surely quite susceptible to loss.

First of all we would like to thank all three reviewers for reading our manuscript so carefully and for their high quality comments and their recommendations. We really appreciate the time you spend on our manuscript. Thank you! Please find below our reply to the individual comments.

Reviewers' comments:

Reviewer #1 (Remarks to the Author):

Overall, these are interesting results and the manuscript itself is generally of high quality. There are a few key changes and additions that need to be made to improve the clarity and reproducibility of the results and several minor comments and corrections, but I suspect these will all be able to be successfully addressed.

However, overall I'm not sure about how novel and influential the presented results are. Firstly, it seems to me that the theoretical extensions to the CV case are impressive without being groundbreaking. Given the existence of previous technical breakthrough's in Ref's 44 and 53 (by some of the same authors as here) which extended the composable security framework, conditional smooth min-entropies, and the derivation of several entropic uncertainty relations to the infinite-dimensional case, it is a valuable, but relatively straightforward, task to derive an CV version of an uncertainty relation in the style of Ref. 42 which in combination with the strong converse of Ref. 32 allows one to follow the analysis of Refs. 20 and 28.

That might be alright if this then allowed an experimental demonstration of noticeably greater performance, but overall it's not clear that this is the case. I should emphasise that this is not due to a poor quality experimental implementation (to the contrary, the EPR resource used here is, to my knowledge, virtually the best in the world) but rather due to in-principal limitations. However, this leads the reader to conclude that, with these current techniques, unclear if there is much benefit in carrying out CV-OT protocols. A more detailed version of these concerns follows below, along with some important substantive questions/criticisms and finally some minor comments.

Continuous-variable technology offers several advantages in comparison to discrete variable technology and deserves a full exploration. Since the same modulation technique and detection is used in the existent telecom environment, it is simpler to integrate. A fact which is acknowledged for example by the telecommunication company Huawei which invests heavily in CVQKD.

Classical noise sources like Brillouin scattering in fibers or scattering and daylight in an atmospheric channel are suppressed by the native filtering of the local oscillator in homodyne detection. We admit that using squeezed states is still some technological hurdle, however, we believe that future work will allow to use coherent states which will drastically simplify the source and in fact will boil it down to the same components that are used in a telecom system. And possibly a future telecom system which takes more care of laser and detection noise could be

directly employed to perform oblivious transfer, QKD and data transmission. Our work is in that sense a milestone towards these future possibilities.

Security proofs for continuous variable quantum cryptography protocols in general are notoriously difficult. This is particularly true in comparison to discrete variables. While it is true that we use some techniques also employed in the discrete case (for example, the reduction of security to the study of strong converses), we respectfully disagree that the analysis is straightforward “in the style of [42]”: entirely new entropic uncertainty relations were derived using very different techniques than were employed than in the discrete case of [42], and indeed any of the other papers dealing with the noisy-storage model (e.g. the use of majorization techniques). We have cited [42] since we relate Renyi entropies to the smooth min entropy, but we would like to point out that this trick is employed in many papers, and is also not the difficult ingredient in this proof: the difficulty then lies in bounding the appropriate Renyi entropy where we use entirely new techniques.

Our work opens the door to further study of such protocols in the domain of CV, where no results were known previously. Issues in our security proof like the 50% loss limit are not fundamental but related to the employed state-of-the-art proof technique and it is an open question how to improve the rates circumventing these issues.

We added the following to the introduction:

“The similarity to classical telecom systems, room temperature operation and intrinsic noise filtering by the local oscillator of homodyne detection will allow seamless integration into telecom networks using wavelength division multiplexing to transmit data and perform oblivious transfer or other quantum cryptographic protocols on the same fiber.”

Furthermore we changed the following in the conclusion:

“Secondly, it is important to derive tight uncertainty relations that hold without additional assumptions. Having such a relation would remove the current constrained on the encoding operation into the quantum memory and possibly also remove the 50 % loss limit.”

%% Comparative weakness in performance %%

a) Strength of assumptions on the memory. To get the best results that they report, the authors need to introduce additional assumptions regarding the encoding operation into the noisy memory. They consider both the case where Bob can only act in an iid fashion on blocks of a fixed length and also the case of Gaussian encoding, with the latter assumption key for achieving the best results. The authors are very up-front about this and the issue is well explained. However, whilst the iid assumption seems conceptually similar to noisy storage itself (essentially a limit on the “depth” of encoding operations) the Gaussian assumptions seems a lot more restrictive given that there are already experimental demonstrations of reasonably high-quality non-Gaussian operations. A lot of these are non-deterministic but in this paper the authors consider a storage rate ν

(which would include non-deterministic encoding operations) between .1 and 1% so there are potentially already encoding operations available today that would violate the Gaussian assumption. Of course it's not at all clear that non-Gaussian encodings would actually prove beneficial in the end, but until the question is settled one way or the other this seems to be a non-trivial extra assumption in comparison to DV-OT.

The referee makes a good point here. As he/she points out we indeed very much like to emphasize this point ourselves. We remark that we operate in this model where security relies on a technological assumption: we assume that it is difficult to store a large number of transmitted signals, and then send more signals than can reliably stored to ensure security. Making additional assumptions is evidently not ideal, but nevertheless meaningful when considering technological restrictions - especially when one takes a practical stance on security and considers not just the technological difficulty but also the cost/time of implementing a particular attack.

It is evidently true that the Gaussian assumption appears more restrictive. We remark that non Gaussian operations that are not deterministic do not confer full cheating power: if the operation fails, an attacker already loses information. In this scenario, an attacker has to encode unknown quantum information, that he/she is unable to prepare him/herself. Hence any failure during the encoding counts as additional noise in the memory. Evidently, our work raises the interesting open question to tighten the analysis, and determine how much power the attacker can really gain by non Gaussian as opposed to Gaussian operations.

If the non-Gaussian attack is carried out on a few qubits in an iid fashion, then such an attack is covered by our iid proof. Current non-deterministic non-Gaussian operations are often limited to a small number of modes so that the iid assumption is not too restrictive.

Similar to quantum key distribution (QKD), a security proof against general (coherent) attacks seems very challenging. Even in the much more established research field of QKD, only security for short distances and in the asymptotic limit is known.

b) The 50% loss limit. This also seems like a significant comparative weakness, although there isn't actually a clear comparison. Starting on line 360 the authors explain that to achieve any secure OT the channel transmission between Alice and Bob must be above 50% even if all other aspects of the implementation are perfect. From my reading of Ref. 24 the same restriction does not hold there. That said, there is an overall tradeoff in Ref.24 between the transmission and the error rate, but my understanding is that at least for an otherwise idealised implementation the transmittance can be arbitrarily low. On the other hand, for many OT applications such as identification I suppose there are operational scenario's such as ATM's where short distance, low loss regime are the relevant ones? Or perhaps for practical non-idealised settings CV-OT is still at a

comparative advantage? I would point out that, whilst the comparison to [24] itself is shows higher rates, Ref. 24 itself was a quite high loss implementation. For instance in a state-of-the-art table top experiment today could have much higher detection efficiencies than 48% that appear there. In any event, the authors should explain the relative CV-DV loss tolerance in more detail.

The 50% loss limit in our implementation comes from the employed analysis and is not a fundamental property of CV oblivious transfer. In Ref 24, losses were separated from other source of noise by a backreporting step. It remains an open question how this could be done here, in conjunction with the uncertainty relations employed..

Having said this we believe that as the reviewer suggests our implementation has still practical applications in short-range scenarios. ATMs are one of those examples, short free-space links in general are another, but also in fiber OT could be implemented within a city's business district with distances of about 3-4km.

Additions in the manuscript:

“However, our implementation is susceptible to losses and requires the optical loss to be generally less than 50%. This limit is a consequence of the analysis we employ in the security proof and is not a fundamental property of CV oblivious transfer.”

“This allows for an implementation of the protocol in short-range applications like a short free-space link with high collection efficiency, e.g. at an ATM, or a short fiber link of maybe 3-4 km within a business district of a city.”

c) Possibly, the status of the strong converse theorem that is utilised (see comment e) but I think this is actually not a problem.

Please refer to the answer to comment (e) below.

%% Major comments %%

d) Insufficient details about the experiment to allow for reproduction of results. In particular the observed covariance matrix is not reported anywhere in the paper, which makes it impossible to attempt to reproduce the curves given in Fig. 3 b). [...]

We followed the suggestion of the reviewer and added the reconstructed covariance matrix to the Method section. We also commented on the detector range.

e) The wording regarding the strong converse for lossy bosonic channels is ambiguous at times. As explained in the paper, to get decent rates a strong converse is necessary to ensure exponential failure of the memory when trying to store information above the classical rate. Also it is explained that Ref. 32 has shown a strong converse for a lossy bosonic channel, provided all of the codewords are contained with high probability in a finite photon-number subspace. However in line 289 and later in 658 the authors write “we will assume that the reliable transmission of classical information over the channel decays exponentially above the classical capacity”. But, if I understand the discussion beginning in line 331, then this assumption is in fact definitely true for this protocol because it satisfies the requirements of Ref 32. Specifically, for the source used although the codeword states do not themselves have a definite photon number, their support lies almost completely within some photon subspace with $n < N_{\max}$. The authors should state this explicitly and (as mentioned above) quote the measured value of N_{\max} determined from the source calibration.

In line 289 and 658 we derive a general theorem that holds for any memory channel (not only bosonic channels) that fulfill the strong-converse property. In line 331 we apply the general theorem to the special case of a bosonic channel, and in order to show that we can apply the general formula in (2) we have to check that a bosonic channel actually satisfies this assumption.

Regarding N_{\max} , N_{\max} are the number of photons encountered in the encoding operation and it is as such an assumption on the quantum memory and not measurable. For completeness, the mean number of photons in the thermal state sent to Bob is about 12, while we assume $N_{\max} = 100$.

We made the following changes in the manuscript:

Caption of figure 2 and 3: “and Bob’s maximal photon number in the encoding *is assumed to be smaller than 100*”

And in the main text: “We computed the security under the Gaussian assumption *and under the assumption on Bob’s quantum memories that the maximal photon number in the encoding is smaller than 100.*”

f) The choice of the α_{cut} parameter. As explained in the paper, any given security parameter ϵ_A fixes a strict lower bound for α_{cut} , i.e. it must be the case that the detectors used are sensitive out to that value and that the source is calibrated such that any value outside that range only occurs with a probability $\epsilon_{\{\alpha_{\text{cut}}\}}$. Currently, the paper provide neither the assumptions nor the experimental data to used to find this lower bound. By rearranging the formula’s given on page 9. I calculate the expression for the minimum allowable value for the detection

probability to be $p^n_{\{\alpha_{\text{cut}}\}} > 1 - \epsilon_A^{2/32}$. If one assumes the quadrature measurements are well approximated by a Gaussian of variance V then one has $p_{\{\alpha_{\text{cut}}\}} = \text{erf}(\alpha_{\text{cut}}/\sqrt{2V})$ which gives $\alpha_{\text{cut}} > \sqrt{2V} \text{erfinverse}[(1 - \epsilon_A^{2/32})^{(1/n)}]$. Taking the worst case value from [27] of $V = 26$ if calculate the lower bounds to be 48 for $n = 10^5$ and 51.8 for $n=10^8$. So it seems that the value used in this paper is okay as long as the data is indeed similar to [27]. But the authors should make this calculation more explicit and, as above, provide variances to allow the reader to check this condition on α_{cut} is satisfied.

We added the calculation to the methods section, yielding $\alpha_{\text{cut}} = 47.9$ for $n=2 \cdot 10^5$, $\epsilon_{\alpha} = 10^{-7}$ and $V=25$ of Alice's state. The chosen value of $\alpha_{\text{cut}}=51.2$ therefore fulfills the security condition.

g) It would be good if there could be some commentary on the optimality or otherwise of the experimental implementation. Specifically, is it actually a practically better strategy to mix two squeezed states to make an EPR state as opposed to a prepare and measure protocol where Alice randomly sends either phase or amplitude squeezed states? And are the squeezing values used in this experiment optimal? In CVQKD situations it is often optimal to use low modulation variance (or low squeezing) in high loss scenarios.

While it is not necessarily better to produce an EPR state rather than a prepare-and-measure protocol the EPR state has practical advantages in the implementation. In a prepare-and-measure scenario Alice has to randomly prepare displaced amplitude or phase squeezed states. In the generation amplitude squeezed states are usually preferred due to the deamplification of technical laser noise of a bright control beam rather than amplification of the noise which can deteriorate the squeezing. Thus, the preferred way to produce either amplitude or phase squeezed beams would be a phase shifter in the squeezed path. However, free-space phase shifters based on piezos are usually slow and a speed of 100 kHz would not be able to achieve. On the other hand phase modulators usually require voltages in the order of kV which have to be driven at high speed. While this is solved in fiber-based phase shifters they usually exhibit 2.5-3dB of loss. Additionally a random displacement has to be implemented. While certainly possible this will introduce another (small) amount of loss and usually introduces another issue in security proofs due to the coarse graining by digital-to-analog converters which only approximates a Gaussian distribution. The generation of EPR states in contrast does not have all these technical problems.

Indeed our experiment is close to optimal with regards to the squeezing degree in the experimentally investigated region of up to 15 % channel loss. Only for channel loss values very close to the maximum possible value the optimal value becomes relevant, though still around 10 dB squeezing. All of this is shown in the newly added Figure 5 in the Methods section which shows the dependence of the oblivious transfer rate on the amount of squeezing used to prepare the EPR state. Additionally we added the following sentence to the main text: "While the

employed EPR entangled state was close to optimal in the investigated regime of up to 15 % channel loss, security can also be obtained with weaker entangled sources, e.g. for a quantum memory storage rate of $\nu=0.001$ only about 4 dB of squeezing are necessary to obtain security for 15 % loss.” and we refer to the Methods section.

h) In the initial discussion the authors characterise the memory in terms of the time Δt that Alice makes Bob wait. But in the actual calculations it is characterised simply in terms of the effective loss of the channel. There doesn't seem to be any discussion of roughly how long Δt should be in order to correspond to a certain loss in realistic quantum memories.

The waiting time Δt sets a certain requirement on the storage time of the quantum memory as malicious Bob has to store *all* of the quantum states for at least this time. In fact the storage time is larger than Δt due to the necessary transmission time for a certain number of quantum states filling up the size of the quantum memory. We furthermore remark that in order to break the protocol one needs a memory that could store all the transmitted states simultaneously.

Typical storage times of quantum memories are milliseconds to seconds with some going up to minutes (<https://arxiv.org/abs/1511.04018>), although we emphasize that there is no data on how well one might store a large number of transmissions simultaneously using a coherent encoding. For an iid attack, of course having very many of these memories, is feasible although evidently costly. For a certain storage time a quantum memory exhibits a certain efficiency which we included in our model. The waiting time is of course linked to the efficiency in the sense that the efficiency becomes smaller for storage times exceeding the nominal storage time of a quantum memory.

In the manuscript we added: “Typical storage times of state-of-the-art quantum memories are milliseconds to seconds with some going up to minutes~\cite{Heshami2016}.” after the sentence: “The classical capacity is determined by the efficiency of the quantum memory for writing, storing (over time Δt) and reading out.”

We note that in our experiment we performed offline rather than online post-processing. When performing online post-processing we would choose Δt such that all the quantum information in state-of-the-art memories would have decayed.

Minor comments

i) Above, I listed information that must be included in order to meet basic standards of reproducibility. In addition, given that it is usually a quite lengthy process to actually calculate rates from expressions like Eq. (2) it would be great to see the codes that were used to evaluate the expressions included as supplementary material. I wouldn't insist

on this but for purposes of transparency and reproducibility I'm sure it would be a very valuable addition.

We uploaded the codes to <https://github.com/qpit/ObliviousTransfer>

Please note that due to lack of time the documentation of the Mathematica scripts is not perfect.

We hope that the notebooks are despite this lack useful.

j) Line 392 the Section number is missing

k) line 401 “bit per symbol” should be “bits per symbol”.

l) line 419 “taken experimental data” should read “experimental data taken”

m) line 591 “lose” should be “loose”

We fixed items (j) to (m). Thank you for finding those mistakes.

n) line 614, its probably a bad idea to use η symbol again given it is already representing loss.

We replaced the symbol by Γ .

Reviewer #2 (Remarks to the Author):

In the submission, an experimental demonstration of a quantum cryptographic protocol, namely 1-out-of-2 oblivious transfer (OT), is performed under the noisy storage model for continuous variable. While I find the work scientifically interesting, I have some questions/comments about it. Therefore, I will withhold my judgement until I see a revised version of the paper. My detailed comments are as follows:

Main Comments:

1) Motivation:

As noted in the Introduction of the paper, the protocol oblivious transfer has already been demonstrated in previous experiments in Refs. [19,24] for the case of discrete variable. In view of these demonstrations, why would yet another demonstration be of broad interest and scientifically important?

Therefore, I suggest that the authors to provide a clear discussion about the motivation of their work. Why is it interesting and important to employ continuous variable for the specific application of 1-out-of-2 OT under the noisy storage model?

Continuous-variable systems offer various advantages over discrete variable systems and deserves full exploration. They share the same encoding operation as classical telecom systems and indeed can use the same components. In our case the detectors are only lower noise versions of detectors used in telecom systems and while the source is so far an entanglement source future implementations and security proofs might consider coherent states. In this sense our work is ground breaking as it opens the possibility to implement oblivious transfer in the same setting as a telecom system. The extension to coherent states which will then allow to use the same source as a telecom system is left for future investigation.

The better integrability of continuous variable systems is acknowledged for example by the telecommunication company Huawei which invests heavily in CVQKD. Other advantages of continuous variables are room temperature operation and intrinsic noise filtering by the local oscillator of the homodyne detection which is important for wavelength division multiplexing of classical data transfer and quantum protocols.

To discuss this we added the following to the introduction:

“The similarity to classical telecom systems, room temperature operation and intrinsic noise filtering by the local oscillator of homodyne detection will allow seamless integration into telecom networks using wavelength division multiplexing to transmit data and perform oblivious transfer or other quantum cryptographic protocols on the same fiber.”

2) Differentiating between distance and loss:

In the second paragraph of column 2 of p. 6, it states that “For short distances (i.e., low channel losses),” . Here, the statement seems to equate channel losses with distances. In my opinion, such a statement is confusing because it completely ignores coupling losses in optical components that are a fact of life and are inevitable in any fiber-based implementation such as the one in Ref. [24]. In fact, a single fiber-optics component such as a coupler could easily give a 1dB or so loss that would render a major modification to the secure OT rate in a continuous variable implementation like the submission.

In summary, my view is that CV implementation of 1-out-of-two OT is only feasible in a low-loss regime. And, such a low loss regime is only realistic in a very short-distance open air setting where various focusing techniques including lenses could be used to increase in the collection efficiency. However, CV implementation of 1-out-of-two OT is unrealistic in a fiber-based setting because even a single fiber-optics component could easily produce a big loss (say 1dB) to the channel that would substantially lower the rate of secure 1-out-of-two OT. Perhaps, these limitations need to be discussed in the paper.

We replaced all mentionings of “short distance” with “low channel loss” as in fact various parameters can influence the possible distance. We agree that the use of fiber components like beam splitters is detrimental. Apart from a possible application in a short free-space link, e.g. at an ATM, we believe however that CV 1-2 OT can be implemented using fibers to distribute the quantum states. Coupling of squeezed beams into AR coated fibers has been shown to exceed 95% efficiency by various groups (Andersen, Furusawa, Schnabel). Together with an efficient outcoupling and free-space homodyne detection a fiber channel with a loss of about 1 dB should be possible which could correspond to 3-4 km of deployed fiber (with a loss of 0.3 dB/km). This makes it possible to use our protocol within a small city or a business district.

We added the following to the paragraph discussing the performance of the protocol:
“This allows for an implementation of the protocol in short-range applications like a short free-space link with high collection efficiency, e.g. at an ATM, or a short fiber link of maybe 3-4 km within a business district of a city.”

3) Fair Comparison with previous results:

It is stated in the submission that “For short distances (i.e., low channel losses), we obtain rates that are about a magnitude larger than those achieved in the DV case [24] while using significantly smaller block sizes of about 10^5 compared to 10^7 .”

I note that the DV implementation [24] was done with fiber-optics components which introduce substantial losses (e.g. order 1dB loss per component). The success of Ref. [24] under such high loss demonstrates the robustness of DV implementation, compared to CV implementation (for the case of 1-out-of-2 oblivious transfer (OT) under the noisy storage model). With such a high loss, CV implementation in the current submission would probably have given a zero rate.

Another point to note is that it is well known that while information reconciliation can be efficiently done for DV QKD, the construction of an efficient method for information reconciliation that takes finite size effects into account remains a major challenge for many CV QKD protocols. For this reason, I am not sure that a comparison on block size only completely captures the demanding situation of CV implementations.

We agree that losses in the DV implementation were quite high and that these would have given zero rate with our implementation. However, looking at Figure 3 of Ref. [24] the source and the detectors were both implemented in free-space optics and only the channel was fiber. In CV detectors with very high quantum efficiency are readily available. Fiber coupling, another huge loss in Ref. [24], has been demonstrated with up to 95% efficiency for CV squeezed and entangled states generated in a cavity. The channel between Alice and Bob itself had in Ref. [24] about 68% efficiency, which just below what our protocol can tolerate. A fair comparison is of course difficult, in particular as in Ref. 24 the intrinsic error rate of the system is of huge importance, a parameter that cannot easily be described here. However, when having a look at Figure 2a the red line (achieved for the measured error rate) indicates about a factor of 3 less rate in comparison to our implementation for about 10% of loss.

We agree that information reconciliation is a major challenge in CV quantum cryptography. While the block size indeed does not completely capture the demanding situation of CV implementations, lower block sizes are for instance preferable for real-time high-speed implementations in FPGAs which usually have memory and bandwidth constraints.

We furthermore note that an improved analysis may lead to CV schemes with a higher loss tolerance (see also comments to ref 1, the 50% loss limit)

Other comments:

4) In the last line of the second column of p. 1, it states that “We overcome this problem by assuming that the dishonest party’s storage operation is Gaussian.”

This seems like a rather strong assumption. I wonder if the authors could provide some justification to such an assumption or discuss about its limitations.

See comment to first referee above, under a) Strength of assumptions on the memory.

5) In line 205, I wonder if the authors could state clearly what value of ϵ_C they are trying to achieve in the present submission and why.

The correctness parameter we were going to achieve is $\sim 10^{-3}$. The limiting factor here is the epsilon for the correctness of the information reconciliation. While we believe it is in fact smaller than the stated value, we deduce this value from the measured data and the fact that the frame

error rate was 0 for 10^3 frames (see Table 1). This is described in detail in the experimental section, “Experimental demonstration of 1-2 rOT”, below Table 1.

6) In line 353-356 “The weakest requirements on the parameters have to be imposed under the Gaussian assumption in which security can already be obtained for low numbers of signals n \approximately 10^5 .” Perhaps, it will be useful to provide a reference here.

Also, is such a conclusion only valid under the Gaussian assumption? Please discuss about the cases of no assumption and i.i.d. assumption.

We refer now to the methods section regarding the assumptions.

For the case of no assumption and i.i.d. assumption about 10^8 signals are required which can be seen (indirectly) from Figure 2 where the curves for these two assumptions are plotted for this value and the one for Gaussian assumption for 10^5 signals. We added this number to the sentence below the quoted one.

7) In line 360-362, it states that “In general, to obtain security a transmittance of the channel between Alice and Bob larger than 0.5 and non-trivial squeezing is required.”

Perhaps, it would be useful for the authors to discuss in practice how hard it is to achieve such a low loss in experimental implementation. Also, explore how much squeezing is required and how hard it is to achieve such an amount of squeezing in a practical setting.

We explore the dependence of the OT rate on the amount of squeezing in the newly added Figure 5 in the Methods section. The necessary amount of squeezing clearly depends on the amount of channel loss and the storage rate of the quantum memories (and the information reconciliation efficiency). In the low loss regime which we studied experimentally security can be obtained for 3-4 dB of squeezing. Only when reaching the upper possible limit for the transmission in the channel high amounts of squeezing are strictly required. We note that for the low loss regime the applied values of 10 dB of squeezing are close to optimal.

10 dB of squeezing are today regularly generated in quantum optics labs and in 24/7 operation in the gravitational wave detector GEO600. We would not expect a large degradation of performance when placing such a squeezer in a 19” rack housing.

8) Starting from line 367, an experimental implementation is presented. However, I find the description lacking as a number of important details such as wavelength and distance are missing.

The wavelength was 1550 nm which is stated in line 378 of the revised manuscript. We introduced losses with a $\lambda/2$, polarizing beam splitter configuration simulating a free-space channel which is described in lines 386ff. The amount of introduced loss can be found in Figure 3b and Table 1. In the conclusion we added the following sentence: “This allows for an

implementation of the protocol in short-range applications like a short free-space link with high collection efficiency, e.g. at an ATM, or a short fiber link of maybe 3-4 km within a business district of a city.”

9) In lines 384-386, it states that “Optical loss in this channel was simulated by a variable beam splitter comprising a half-wave plate and a polarizing beam splitter.”

I don’t understand this statement. Wasn’t it a real experiment that your team has performed? How do you do the simulation here?

Using the word “simulated” was a mistake. We indeed meant “introduced” instead which we have now changed in the revised manuscript.

10) In lines 396-399, it states that “We then chose from each set the first 10^5 for post-processing (i.e., $n = 2 \times 10^5$) to keep the block size of the information reconciliation code constant.”

Doesn’t this violate the fair sampling assumption? Why is the protocol secure?

Please note that there is no sampling (of the error rate nor anything else) going on here. We simply set the block size for an individual protocol run to be $n=2 \times 10^5$, limiting ourselves to take only $n=2 \times 10^5$ signals of each protocol run (with 2.03×10^5 signals each) into account. This cropping of the initial data does not have any impact on security, because the choice of n is a fixed public protocol parameter that no dishonest player has control over.

The fact that symbols close to the end of the initial block of size 2.03×10^5 have a smaller probability to end up in any of the two sets I_0 and I_1 of size 10^5 does not change the security of the protocol. The dishonest partner cannot influence which symbols will end up in which set and which symbols will not be used if the other player follows the protocol and chooses his bases independent and identically distributed both with probability $\frac{1}{2}$.

11) On line 418, it states that $\epsilon_{IR} = 10^{-3}$. This seems quite large (compared to standard QKD security parameters of about 10^{-10}). Is there any reason for such a large number?

It is indeed quite large in comparison to standard QKD numbers. The reason is that it is not simple to validate the correctness of the information reconciliation protocol to such an extent as this can only be done empirically and not analytically for LDPC. This is in contrast to QKD where failures of frames can be detected with probability (almost) arbitrarily close to one by exchanging universal hashing tags and usually be handled without abortion of the protocol. In OT, a failure of the information reconciliation will stay unnoticed because of the fact that partners can cheat rules out a similar confirmation step. We note that we here achieve similar values as in Ref. 24.

12) On line 430, it says “the lossless channel”. Why consider only a lossless channel?

In the protocol the covariance matrix of the state Alice sends to Bob has to be estimated. Since our experimental setup allowed us to control the loss in the channel we used the setting of “zero” loss to characterize the source instead of removing the channel and setting up a new homodyne detector just at the output of Alice. So the only purpose of the lossless channel is for characterization.

13) On line 430, it says that “We see that for short distances...” As mentioned in one of the Main Comments above, I think it is important to differentiate clearly between short distances and losses and state clearly if one is considering free-space or fiber-based setting.

Yes, we agree with the reviewer and have changed the phrase to use loss instead of distance. We amended this throughout the manuscript except one paragraph where we compare the possible channel loss with a scenario in optical fibers.

14) On line 435, are the numbers “ $\nu=0.01$ ” and “ $\nu=0.001$ ” given there realistic? Why?

For $n = 2 \cdot 10^5$ transmitted quantum states, a storage rate of 0.001 implies that malicious Bob implements 200 working quantum memories. From an experimental point of view this is quite challenging today and an even lower number might be more realistic, however, we intend the protocol to be near-future proof which is the reason why we chose these two more conservative numbers.

15) On lines 439-440, it states that “showed security against a malicious party with an imperfect quantum storage device.” Perhaps, you mean “highly imperfect” quantum storage devices because there is severe limit on the rate of OT due to the imperfections of quantum storage devices.

We agree with the reviewer and added the word “highly” to the sentence.

16) On lines 444-446, “However, security can also been obtained with weaker entangled sources than used in our experiment such that on chip entanglement sources are possible [39].” If one includes coupling loss or collection efficiency loss into consideration, I am not sure if the above statement is still correct. Any thought?

This is in fact a good point. The coupling efficiency in Ref. 39 was 72% which basically exceeds our channel loss margin. We removed the reference to the on-chip entanglement paper and instead mention that only about 4 dB of squeezing are necessary to obtain security: “While the employed EPR entangled state was close to optimal in the investigated regime of up to 15 % channel loss, security can also be obtained with weaker entangled sources, e.g. for a quantum

memory storage rate of $\nu=0.001$ only about 4 dB of squeezing are necessary to obtain security for 15 % loss.” This is backed up by a new figure introduced to the Methods section showing the dependence of the oblivious transfer rate under Gaussian assumption versus the amount of squeezing used to generate the EPR state.

Reviewer #3 (Remarks to the Author):

The paper is well written and the results are scientifically sound. It is a very interesting and promising approach as a continuous-variable (CV) platform is highly compatible with real world telecommunication systems and relatively easy to handle. Also the sending rates can be quite high, which together with high quantum efficiency of homodyne detection used makes the overall system very effective. The challenging part is the security proof; the approaches from the discrete variables (DV) protocols cannot be easily translated over. The work is quite unique as oblivious transfer is one of the most important cryptographic primitives and this is I believe the first experimental implementation of such types of two-party protocols in CV regime. Also in DV regime, the only experimental implementation, also using noisy-storage model, has been performed quite recently, in 2014, and a bit earlier a bit-commitment protocol, in the same model. This current paper presents a neat experimental work and an advanced novel security proof, of importance both for the fundamental understanding of secure quantum protocols and for applications in quantum technologies. I fully recommend the publication in Nature Communications. I have a few comments though.

The security proof is based on new entropic uncertainty relation for the field quadratures, or more precisely on founding a good bound for the smooth entropy. The equation (1) is an important result, not only for the OT protocol, many CV secure protocols rely on properly bounding smooth-min entropy for CV platform, which is still largely an open question. It would be very helpful to have a comparison with the state-of-art and a comment on applicability of the derived inequality beyond this particular protocol.

We agree with the referee that our techniques and the bound on the smooth min-entropy might be valuable for other applications as well. To the best of our knowledge, our bound and the majorization technique by Rudnicki has so far not been used in other contexts, and it is an interesting open question to obtain a tighter bound by taking into account more of the constraints.

The inequality can also be applied to prove security of other two party protocols in the noisy storage model such as bit commitment or secure password based identification. It is important to remark that the uncertainty relation is fundamentally different to the one usually used in quantum key distribution.

I do not quite understand why it is more convenient to use the two-mode squeezed vacuum rather than doing a prepare-and-measure scheme. Would switching to prepare-and-measure scheme relax the loss limit? The EPR states are surely quite susceptible to loss.

The entanglement based version and a prepare-and-measure-scheme using squeezed vacuum states are equivalent, i.e. both have the same susceptibility to loss and exhibit the same loss limit.

While it is not necessarily better to produce an EPR state rather than implement a prepare-and-measure protocol, the EPR state has practical advantages in the implementation. In a prepare-and-measure scenario Alice has to randomly prepare displaced amplitude or phase squeezed states. In the generation amplitude squeezed states are usually preferred due to the deamplification of technical laser noise of a bright control beam rather than amplification of the noise which can deteriorate the squeezing. Thus, the preferred way to produce either amplitude or phase squeezed beams would be a phase shifter in the squeezed path. However, free-space phase shifters based on piezos are usually slow and we would not be able to achieve a speed of 100 kHz or even higher values in future implementations. On the other hand bulk phase modulators usually require voltages in the order of kV which have to be driven at high speed. While this is solved in fiber-based phase shifters they usually exhibit 2.5-3dB of loss. Additionally a random displacement has to be implemented. While certainly possible this will introduce another (small) amount of loss and usually introduces another issue in security proofs due to the coarse graining by digital-to-analog converters which only approximates a Gaussian distribution. The generation of EPR states in contrast does not have all these technical problems.

Another advantage of EPR states is related to trojan horse attacks. A malicious Bob could try to hack into Alice's source by launching a trojan horse attack on the modulator used to prepare the displaced squeezed states. EPR states in contrast are intrinsically immune against such attacks (Lo, H.-K. & Chau, H. F. Unconditional security of quantum key distribution over arbitrarily long distances. *Science* 283, 2050–2056 (1999)).

Reviewers' comments:

Reviewer #1 (Remarks to the Author):

Overall, the authors have done a good job responding my comments and made some good points regarding the novelty and importance of their paper. Upon reflection, I take their point that while derivation of the uncertainty relations uses tools from [42] and the continuous variable works of [20] and [28] there are several features, particularly the majorisation approach that are not contained in either of these works. Similarly although the necessity of the Gaussian assumption and the 50% loss limit are not ideal there is no reason to assume either drawback is incapable of being remedied if further work. For the moment it seems important to demonstrate the possibility of these protocols in the CV regime and indeed it seems clear that in two-party cryptography there are several practical applications over very short distances where the low loss tolerance but high rates of CV systems tend to do well. They have also added several more details which make it possible to reproduce the calculations quite swiftly (the uploaded codes are also extremely helpful in this regard). In light of this I'm now satisfied that this work is suitable for publication in Nature Communications. However, I have found a few other minor errors/queries which I list below.

%%% Minor Comments %%%

Line 16: "allows to establish security" should read "allows us to establish security" or "establishes security"

In equation (4) I think $p_{\{k_j\}}$ should just be p_j

Line 584 I don't think the quantity $c(\cdot)$ hasn't been defined yet. In any event as far as I can see this quantity is already expressed by the function $\gamma()$

Also, the summations that appear in lines 556, 577 and others aren't specified but I gather they're infinite.

Line 611: the expression $g(x)$ appears to be missing a factor of $1/\sqrt{\alpha}$

Line 735: "measure" should be "measured"

Reviewer #2 (Remarks to the Author):

I thank the authors for their revised manuscript and their helpful reply. While I find the paper scientifically interesting, just like the first referee, I do not find it groundbreaking. This is partly because the current protocol has severe limit on channel loss (it must be less than 50%) and the experimental demonstration provided is only proof-of-principle that does not take many real-life practical issues (such as limited collection efficiency of telescopes or coupling loss) into account. Therefore, I find that the claim of the practicality of applications to several km of optical fibers a bit too strong. I think the paper is more suitable for a specialized journal. My detailed comments are as follows:

1) Motivation:

Please note that I do not doubt the importance of CV-QKD itself. My question concerns specifically the protocol oblivious transfer (OT). For OT based on noisy-storage model, at least for the current theory discussed in the submission, using continuous variable (CV) will severely limit the channel loss to less than 50%. This is a rather severe restriction that will limit substantially not only the distance, but also the practicality of CV-based OT protocol. This issue will be discussed further in later comments here.

In this sense, at least for the moment, I think it is more natural for people to use discrete-variable (DV) for OT. And, CV does not seem to offer much advantage over DV for OT applications. Therefore, the motivation of the paper is a bit unclear to me.

Another point to note is that for quantum communication protocols such as OT, the homodyne detector used must be low-noise and perhaps broad-band. These stringent requirements make the design and building of homodyne detectors for quantum communication purposes rather challenging. In summary, one could not just take a random homodyne detector used in conventional communication and use it in quantum communication and expect it to work. I think it is important to make this point clear in a manuscript.

2) Restriction of channel loss to 50%.

For the current protocol, it is insecure when the channel loss is 50% or higher. Considering finite-size effects etc, the simulation result in Figure 3 shows that the secure OT bit rate is positive only for the region where the channel loss is at most 25%-30%. Please note that this is a rather small loss for quantum communication systems.

In the reply, the authors mention that focusing lens could increase the collection efficiency. This is true. However, if we are considering free-space applications over 1km or so, then big lens and perhaps telescopes will probably need to be used. Owing to atmospheric turbulence, beam wandering is an issue. So, collection efficiency may well be limited. If, as usually done, one subsequently attempts to couple the collected photon to an optical fiber, then the quantum signal suffers further from the coupling loss from the lens to an optical fiber. The point that I am making is that all those losses add up. So, the secure OT bit rate goes to zero pretty quickly.

3) The experimental demonstration is only proof-of-principle.

The experiment performed is only a proof-of-principle one. The distance in the experiment is actually *zero*. Loss is introduced by a variable beam-splitter comprising a half-wave plate and a polarizing beam splitter. The same detectors are used for characterization. [This essentially assumes that Alice's detectors have exactly the same characteristics as Bob's detectors.] Such a proof-of-principle experiment, while interesting, does not deal with many real-life imperfections in applications. For instance, in real-life, there might be misalignment/drift in the relative measurement bases used by Alice and Bob. The calibration of the detectors used by Alice and Bob would probably be different. In free-space, the beam will wander due to atmospheric turbulence. So, the collection efficiency will be limited and it will also fluctuate with time. The coupling efficiency between various components also contributes to the loss budget. The excess noise of the detectors needs to be taken into account, etc.

In summary, I think in both the Abstract and the main text, it is important to state clearly that the performed experiment is a proof-of-principle one and it does not take fully into account the many imperfections stated above.

4) Basis-dependence and fair sampling assumption

In the authors' reply, it is claimed that "The cropping of this initial data does not have any impact of security..". To be honest, I am not entirely sure if this is true. As pointed by in C. Pfister, N. Lütkenhaus, S. Wehner, and P. J. Coles, *New J. Phys.* 18, 053001 (2016), whenever there is some basis-dependence in the operations by Alice and Bob, security might be in doubt. (See also Kiyoshi Tamaki et al, *Quantum Sci. Technol.* 3, 014002 (2018) for a discussion.) Doesn't cropping of the initial data constitute to a basis-dependent operation that might affect security and violate the fair sampling assumption?

5) Assumptions made

On line 735-736, it states that “where the entries in brackets were not measure, but assumed to be 0”. Why is such an assumption made? Why should trust such an assumption?

Also, the paper makes Gaussian assumption on the encoding. This seems like a rather strong assumption.

On line 435-436, it says that “the maximal photon number in the encoding is smaller than 100”. How is this number verified in the actual experiment? How can one be sure that the intensity of the signal does not fluctuate widely so that some signals might have a huge number of photons? How do the authors propose to measure accurately the photon number in each pulse in a high-speed CV QKD system?

6) Some implementation details seem to be missing.

Some implementation details seem to be missing. For instance, it is unclear to me how the LDPC code is implemented in practice in the experiment. Software? Which programming language? Hardware? FPGA? How was it encoded? What was the speed achieved?

For instance, it says the two-universal hash function were implemented using Toeplitz matrix. But, how exactly was it implemented? What parameters are chosen? etc etc.

Reviewer #1 (Remarks to the Author):

%%% Minor Comments %%%

Line 16: “allows to establish security” should read “allows us to establish security” or “establishes security”

We added “us” to the sentence.

In equation (4) I think $p_{\{k_j\}}$ should just be p_j

Yes, this is correct. We fixed it.

Line 584 I don’t think the quantity $c(\cdot)$ hasn’t been defined yet. In any event as far as I can see this quantity is already expressed by the function $\gamma(\cdot)$

Yes, it should have been gamma.

Also, the summations that appear in lines 556, 577 and others aren’t specified but I gather they’re infinite.

Yes, the summation is over all N (natural number). We added this accordingly.

Line 611: the expression $g(x)$ appears to be missing a factor of $1/\sqrt{\alpha}$

Yes indeed this factor was missing.

Line 735: “measure” should be “measured”

Amended.

Reviewer #2 (Remarks to the Author):

1) Motivation:

Please note that I do not doubt the importance of CV-QKD itself. My question concerns specifically the protocol oblivious transfer (OT). For OT based on noisy-storage model, at least for the current theory discussed in the submission, using continuous variable (CV) will severely limit the channel loss to less than 50%. This is a rather severe restriction that will limit substantially not only the distance, but also the practicality of CV-based OT protocol. This issue will be discussed further in later comments here.

In this sense, at least for the moment, I think it is more natural for people to use discrete-variable (DV) for OT. And, CV does not seem to offer much advantage over DV for OT applications. Therefore, the motivation of the paper is a bit unclear to me.

Quantum communication protocols not only have innovative applications over long distances. Also kilometer-scale quantum networks with highest security-level are extremely important, e.g. for government districts and business premises. On such short distances, loss of less than 50% can easily be achieved with CV protocols in standard telecommunication fiber networks while offering larger rates than discrete-variable protocols.

As the referee points out, below-kilometer-scale free-space links might be unfeasible, very short distance links over a couple of centimeters, however, offer high transmission and are not susceptible for atmospheric disturbances and similar effects. Those very short free-space links are useful for example for an ATM. Oblivious transfer, or rather the derived protocol of password-based identification, offers here additional security, for instance, against a cheating ATM which cannot find out the pin.

Another point is that OT is complete for secure multi-party computation (see Joe Kilian, Founding cryptography on oblivious transfer. In Proceedings of the twentieth annual ACM symposium on Theory of computing, STOC '88, pages 20–31, New York, NY, USA, 1988. ACM and Yuval Ishai, Manoj Prabhakaran, and Amit Sahai. Founding Cryptography on Oblivious Transfer - Efficiently. In CRYPTO '08, pages 572–591, 2008). Thus, OT can be used to construct many protocols, that are (in contrast to key distribution) even useful if all parties *sit together on a table (i.e. have zero distance)* because they prevent the other parties to learn the secret input values.

We further note that the 50% loss limit is not a fundamental issue of CV OT but rather an artifact from our state-of-the-art proof technique. We are convinced that future work will circumvent this current limitation.

In conclusion, we indeed consider our work to be groundbreaking as it demonstrates OT for the first time in the very important setting of continuous-variable systems.

Another point to note is that for quantum communication protocols such as OT, the homodyne detector used must be low-noise and perhaps broad-band. These stringent requirements make the design and building of homodyne detectors for quantum communication purposes rather challenging. In summary, one could not just take a random homodyne detector used in conventional communication and use it in quantum communication and expect it to work. I think it is important to make this point clear in a manuscript.

It is true that the homodyne detector has to be low-noise which is in fact also the case for CV-QKD. Low noise homodyne detectors with a bandwidth of several 10 MHz up to 100 MHz are nowadays rather standard in quantum optical labs. A low-noise, broadband detector (> 1 GHz) has for example been demonstrated in Optics Letters Vol. 41, Issue 21, pp. 5094-5097 (2016).

We now mention this fact in the manuscript where we write:

“Alice kept one of the entangled modes and performed balanced homodyne detection using a low-noise, high quantum efficiency homodyne detector“, and we refer to the methods section for details.

2) Restriction of channel loss to 50%.

For the current protocol, it is insecure when the channel loss is 50% or higher. Considering finite-size effects etc, the simulation result in Figure 3 shows that the secure OT bit rate is positive only for the region where the channel loss is at most 25%-30%. Please note that this is a rather small loss for quantum communication systems.

In the reply, the authors mention that focusing lens could increase the collection efficiency. This is true. However, if we are considering free-space applications over 1km or so, then big lens and perhaps telescopes will probably need to be used. Owing to atmospheric turbulence, beam wandering is an issue. So, collection efficiency may well be limited. If, as usually done, one subsequently attempts to couple the collected photon to an optical fiber, then the quantum signal suffers further from the coupling loss from the lens to an optical fiber. The point that I am making is that all those losses add up. So, the secure OT bit rate goes to zero pretty quickly.

Please note that we are not considering a 1 km free-space channel. We mention that it might be possible to implement it in a very short range free-space application like an ATM, where the collection efficiency can indeed be high and atmospheric turbulence, etc. do not play a role. We also mention also a short fiber link of 3-4 km. Here fiber in-coupling is most critical, but has been demonstrated with 95% efficiency in several labs, e.g. in [PRL 104, 251102 (2010)]. Since homodyne detection is dependent on polarization, it has to be stabilized at the receiver station which is routinely performed in CV QKD. We believe that his scenario is most suitable to implement our oblivious-transfer protocol.

3) The experimental demonstration is only proof-of-principle.

The experiment performed is only a proof-of-principle one. The distance in the experiment is actually *zero*. Loss is introduced by a variable beam-splitter comprising a half-wave plate and a polarizing beam splitter. The same detectors are used for characterization. [This essentially assumes that Alice's detectors have exactly the same characteristics as Bob's detectors.] Such a proof-of-principle experiment, while interesting, does not deal with many real-life imperfections in applications. For instance, in real-life, there might be misalignment/drift in the relative measurement bases used by Alice and Bob. The calibration of the detectors used by Alice and Bob would probably be different. In free-space, the beam will wander due to atmospheric turbulence. So, the collection efficiency will be limited and it will also fluctuate with time. The coupling efficiency between various components also contributes to the loss budget. The excess noise of the detectors needs to be taken into account, etc.

In summary, I think in both the Abstract and the main text, it is important to state clearly that the performed experiment is a proof-of-principle one and it does not take fully into account the many imperfections stated above.

We agree that our experiment is a proof-of-principle experiment in several ways. To make this more clear we now write in the abstract:

“We experimentally demonstrate in a proof-of-principle experiment the proposed oblivious transfer protocol for various channel losses by using entangled two-mode squeezed states measured with balanced homodyne detection.”

And in the main text:

“Here, we propose and experimentally demonstrate in a proof-of-principle experiment an oblivious transfer protocol based on optical continuous-variable (CV) systems.”

4) Basis-dependence and fair sampling assumption

In the authors' reply, it is claimed that “The cropping of this initial data does not have any impact of security..”. To be honest, I am not entirely sure if this is true. As pointed by in C. Pfister, N. Lutkenhaus, S. Wehner, and P. J. Coles, *New J. Phys.* 18, 053001 (2016), whenever there is some basis-dependence in the operations by Alice and Bob, security might be in doubt. (See also Kiyoshi Tamaki et al, *Quantum Sci. Technol.* 3, 014002 (2018) for a discussion.) Doesn't cropping of the initial data constitute to a basis-dependent operation that might affect security and violate the fair sampling assumption?

Notice that unlike in QKD (which the two mentioned articles are about), our two-party protocol *does not have any sampling phase* where an error/interference rate is determined. Therefore, we make no appeal to a fair-sampling assumption.

The fact that in our implementation, we include exactly the first $n/2$ signals of each of the two sets (of signals where Alice and Bob measured in the same or in different bases, respectively) is determined beforehand, and the choice of these sets is out of control of any dishonest player, because the honest player chooses his/her basis string uniformly at random.

We added the following to the manuscript to make this point clear:

“From a security perspective this is possible because the size of the set is determined beforehand as part of the protocol. Because the honest player chooses his/her basis string uniformly at random, the choice of these sets is thus out of control of any dishonest player.”

5) Assumptions made

On line 735-736, it states that “where the entries in brackets were not measure, but assumed to be 0”. Why is such an assumption made? Why should trust such an assumption?

Please note that the security of our protocol is independent of the fully reconstructed covariance matrix of the employed EPR states. The only important parameters are the two variances of Alice’s state since the cut-off parameter depends on it, and the correlation coefficient of Alice’s and Bob’s measurement outcomes (related to the variance and covariance of the same quadratures). The latter is used to pick a suitable LDPC code. The full covariance matrix is not used at all to obtain security! This means that the characterization at the beginning of the protocol can actually be performed by Alice’s and Bob’s real devices. Alice do not need a copy of Bob on her own to determine the state.

To make this point more clear we changed the paragraph describing the parameter estimation before the run of the protocol. We now write:

“Before Alice and Bob start the actual protocol, they estimate the necessary parameters to run the protocol. The EPR source is located in Alice’s lab who is using balanced homodyne detection to estimate the variance of her local thermal state to fix $\alpha_{\text{cut}} > 0$ such that the probability for her to measure a quadrature with an absolute value smaller than α_{cut} is larger than $p_{\alpha_{\text{cut}}}$ ($p_{\alpha_{\text{cut}}} \approx 1$). Alice and Bob then estimate the correlation coefficient of their measurement outcomes, measured jointly in the same quadrature, to choose an appropriate information reconciliation (IR) code for the protocol. We note that this estimate can be made safely before the protocol even if one of the parties later tries to break the security (see [24] for a discussion).”

We note, that the reconstructed covariance matrix as displayed in the methods is only used to obtain theory curves in the various figures. We assumed some entries to be 0 since they usually are very close to zero and it simplified the measurement. The obtained theory curves fit well with the measurement results.

Also, the paper makes Gaussian assumption on the encoding. This seems like a rather strong assumption.

Referee #1 had a similar remark in the first round where he/she pointed out that we need additional assumptions on the encoding operation and that we considered an iid and a Gaussian assumption. Below we quote our answer:

“The referee makes a good point here. As he/she points out we indeed very much like to emphasize this [that we need additional assumptions on the encoding] point ourselves. We remark that we operate in this model where security relies on a technological assumption: we assume that it is difficult to store a large number of transmitted signals, and then send more signals than can reliably stored to ensure security. Making additional assumptions is evidently not ideal, but nevertheless meaningful when considering technological restrictions - especially when one takes a practical stance on security and considers not just the technological difficulty but also the cost/time of implementing a particular attack.

It is evidently true that the Gaussian assumption appears more restrictive. We remark that non Gaussian operations that are not deterministic do not confer full cheating power: if the operation fails, an attacker already loses information. In this scenario, an attacker has to encode unknown quantum information, that he/she is unable to prepare him/herself. Hence any failure during the encoding counts as additional noise in the memory. Evidently, our work raises the interesting open question to tighten the analysis, and determine how much power the attacker can really gain by non Gaussian as opposed to Gaussian operations.

If the non-Gaussian attack is carried out on a few qubits in an iid fashion, then such an attack is covered by our iid proof. Current non-deterministic non-Gaussian operations are often limited to a small number of modes so that the iid assumption is not too restrictive.”

On line 435-436, it says that “the maximal photon number in the encoding is smaller than 100”. How is this number verified in the actual experiment? How can one be sure that the intensity of the signal does not fluctuate widely so that some signals might have a huge number of photons? How do the authors propose to measure accurately the photon number in each pulse in a high-speed CV QKD system?

It is the maximal photon number encountered in the encoding operation and it is as such an assumption on the quantum memory and not measurable. It is independent of the number of photons in the quantum state which is why we do not have to measure the photon number. Please see also our answer in the first round to referee 1.

6) Some implementation details seem to be missing.

Some implementation details seem to be missing. For instance, it is unclear to me how the LDPC code is implemented in practice in the experiment. Software? Which programming language? Hardware? FPGA? How was it encoded? What was the speed achieved?

We added to the methods: “We used C++11 as programming language, compiled with GNU GCC 6.3, and ran the binary on a single core of an Intel Xeon E7-8870v2 CPU in a PC running Linux (Debian 8) as operating system. On average we achieved a rate of approximately 1k oblivious bit transfers per second.”

For instance, it says the two-universal hash function were implemented using Toeplitz matrix. But, how exactly was it implemented? What parameters are chosen? etc etc.

We clarified all those questions in the revised manuscript. In particular we added the following paragraph:

“As family of two-universal hashing functions we selected the mapping of the binary input string to the binary output string by multiplying the input string with a uniformly randomly chosen binary Toeplitz matrix T . Multiplication by a Toeplitz matrix is equivalent to linear cross-correlation. This allowed us to make use of the number-theoretic transform to obtain an implementation with computational complexity $O(n \log n)$ and without floating point errors. The binary input strings had a total length of 10^6 bits (consisting of $n/2 = 10^5$ symbols with 10 bits per symbol).

The binary output strings had a length of $\lfloor \ell \rfloor$. Thus the size of T was $\lfloor \ell \rfloor \times 10^6$. The seed (the values for the first row and first column of the Toeplitz matrix) was generated with the quantum random number generator.”

Furthermore we added the output length for a storage rate of 0.001 to Table 1. These values are displayed in Figure 3 as well.

REVIEWERS' COMMENTS:

Reviewer #2 (Remarks to the Author):

I recommend the publication of the paper after making the following mandatory changes.

---In line 370, replace “an” by “a proof-of-principle”.

[This is to make it clear that the experiment is only proof-of-principle.]

---In line 504, note explicitly that, in making the statement that the protocol is feasible for a short fiber-link of 3-4km, the authors do not consider insertion loss due to standard fiber optics components such as fiber couplers. Nor is splicing loss considered.

[Note that standard fiber optics components can easily have 3dB insertion loss (see e.g. https://www.thorlabs.com/newgrouppage9.cfm?objectgroup_id=6673) which renders the protocol insecure.]

We are happy to hear that referee 2 now also supports our work. Please find below our response to the final comments of the referee.

**In line 370, replace “an” by “a proof-of-principle”.
[This is to make it clear that the experiment is only proof-of-principle.]**

We replaced it as requested.

In line 504, note explicitly that, in making the statement that the protocol is feasible for a short fiber-link of 3-4km, the authors do not consider insertion loss due to standard fiber optics components such as fiber couplers. Nor is splicing loss considered.

[Note that standard fiber optics components can easily have 3dB insertion loss (see e.g. https://www.thorlabs.com/newgrouppage9.cfm?objectgroup_id=6673) which renders the protocol insecure.]

We added a description of the parameters we used to calculate the distance. While the referee is right that fiber optic components can have quite substantial insertion loss, we in fact consider a free-space receiver station so that only the insertion loss into the fiber and the transmission loss of the fiber play a role. For the latter we used a realistic value of 0.3 dB/km instead of the standard value of 0.2 dB/km which is achievable for fiber spools in the lab, but not necessarily in a deployed fiber. We now added:

“Here we assumed a free-space to fiber coupling efficiency of 95% (achievable with anti-reflex coated fibers), a realistic fiber transmission loss of 0.3 dB/km at 1550 nm and a high efficiency free-space homodyne receiver as implemented in our experiment.”